# Engineering broad-spectrum inhibitors of inflammatory chemokines from subclass A3 tick evasins

Shankar Raj Devkota [1], Pramod Aryal [1], Rina Pokhrel[1], Wanting Jiao [2], Andrew Perry [3], Santosh Panjikar [1,4], Richard J. Payne [5,6], Matthew C. J. Wilce[1], Ram Prasad Bhusal [1] & Martin J. Stone [1]

Chemokines are key regulators of leukocyte trafficking and attractive targets for anti-inflammatory therapy. Evasins are chemokine-binding proteins from tick saliva, whose application as anti-inflammatory therapeutics will require manipulation of their chemokine target selectivity. Here we describe subclass A3 evasins, which are unique to the tick genus *Amblyomma* and distinguished from "classical" class A1 evasins by an additional disulfide bond near the chemokine recognition interface. The A3 evasin EVA-AAM1001 (EVA-A) bound to CC chemokines and inhibited their receptor activation. Unlike A1 evasins, EVA-A was not highly dependent on N- and C-terminal regions to differentiate chemokine targets. Structures of chemokine-bound EVA-A revealed a deep hydrophobic pocket, unique to A3 evasins, that interacts with the residue immediately following the CC motif of the chemokine. Mutations to this pocket altered the chemokine selectivity of EVA-A. Thus, class A3 evasins provide a suitable platform for engineering proteins with applications in research, diagnosis or anti-inflammatory therapy.

Inflammation, the body's response to injury or infection, is characterised by the recruitment of leukocytes to the affected tissues. Leukocyte recruitment plays an essential role in immune homeostasis[1] but dysregulation contributes to the progression of numerous immune and inflammatory diseases, including asthma[2], atherosclerosis[3], multiple sclerosis[4] and cancer[5]. Therefore, selective control of leukocyte recruitment is an attractive anti-inflammatory strategy.

Leukocyte recruitment in inflammation is initiated by the interactions between chemokines, soluble proteins expressed in the affected tissues, and chemokine receptors, which are G protein-coupled receptors (GPCRs) expressed on the surfaces of leukocytes[6]. Despite the well-known "druggability" of GPCRs, inhibition of specific

chemokine receptors has largely been unsuccessful as an anti-inflammatory approach[7]. This is, at least in part, because most inflammatory diseases involve the secretion of numerous chemokines, which activate several receptors, often distributed across multiple classes of leukocytes. In this light, there is interest in the alternative strategy of simultaneously inhibiting several chemokines or chemokine receptors. Considering that blockade of multiple chemokines is a survival strategy used by viruses and parasites such as worms and ticks to suppress host inflammatory defences[8], it may be possible to re-purpose the natural proteins of these organisms as clinical anti-inflammatory agents.

The Ixodidae (hard ticks) are ectoparasitic arachnids that survive on the blood of their hosts and are classified anatomically into a single

[1]Monash Biomedicine Discovery Institute, and Department of Biochemistry and Molecular Biology, Monash University, Clayton, VIC 3800, Australia. [2]Ferrier Research Institute, Victoria University of Wellington, Wellington 6140, New Zealand; Maurice Wilkins Centre for Molecular Biodiscovery, Auckland 1142, New Zealand. [3]Monash Bioinformatics Platform, Monash Biomedicine Discovery Institute, Monash University, Clayton, VIC 3800, Australia. [4]Australian Synchrotron, ANSTO, Clayton, VIC 3168, Australia. [5]School of Chemistry, The University of Sydney, Sydney, NSW 2006, Australia. [6]Australian Research Council Centre of Excellence for Innovations in Peptide and Protein Science, The University of Sydney, Sydney, NSW 2006, Australia. ✉e-mail: ram.bhusal@monash.edu; martin.stone@monash.edu

genus of prostriate and several genera of metastriate species. Their host compatibility is acquired via a cocktail of salivary proteins secreted into the bite site, including two families of chemokine-inhibitory proteins called evasins[9]. By inhibiting host chemokines, evasins supress inflammation at the site of the tick bite, which is believed to prolong the period of blood feeding before the host becomes aware of the tick and removes it. Class A and class B evasins are structurally unrelated and specifically inhibit CC and CXC chemokines, respectively, the two major families of chemokines[10]. However, each evasin targets a unique spectrum of chemokines within its cognate family. For example, the class A evasins EVA-1 and EVA-4 bind to four and 17 human CC chemokines, respectively[11], whereas the class B evasin EVA-3 binds to six CXC chemokines[12]. Evasins from both families have exhibited anti-inflammatory activity in animal models of inflammatory diseases[10]. Thus, evasins constitute a rich pool of anti-inflammatory proteins with distinct chemokine selectivity profiles, and potential as human therapeutics.

Developing evasins as effective therapeutics is likely to require modification of natural evasins to target the relevant chemokines in specific inflammatory diseases. To this end, it would be beneficial to identify different classes of evasins and it is essential to understand the molecular features that underpin chemokine recognition. A recent phylogenetic analysis revealed that class A evasins can be subdivided into a major subclass (class A1), which is spread across several metastriate tick genera, and a minor subclass (class A2), which is uniquely expressed in prostriate ticks of the genus *Ixodes*[13].

We now describe a third subclass of class A evasins (designated class A3), which are unique to the metastriate tick genus *Amblyomma*, indicating that they have evolved from class A1 evasins. Structures of a class A3 evasin bound to several chemokines, along with mutational analysis, revealed the chemokine recognition determinants that are either shared with class A1 evasins or unique to the A3 subclass. Moreover, modification of the chemokine interface of the class A3 evasin yielded engineered evasins with modified selectivity amongst CC chemokines. Thus, this study establishes class A3 evasins as a potential source of clinically useful anti-inflammatory proteins.

## Results
### Identification and phylogeny of class A3 evasins
Class A1 evasins contain eight strictly conserved Cys residues (forming four disulfide bonds), whereas class A2 evasins lack one pair of the conserved cysteine residues (Fig. 1a)[10]. We have now identified 14 evasin-like sequences with two additional Cys residues in conserved positions corresponding to the N-terminal region and the fourth β-strand, based on known structures of class A1 evasins (Fig. 1a and Supplementary Fig. 1). Considering that these structural elements are adjacent in evasin three-dimensional structures[14], they have the potential to form a fifth disulfide bond. Moreover, as the N-terminal region was known to be important for chemokine recognition[11,15], we envisaged that this structural feature would likely influence function. In addition to ten conserved Cys residues, these evasin-like sequences contain several other features consistent with them being functional evasins[14,16], including secretion signal sequences, two Gly residues conserved amongst class A1 evasins, several potential N-linked glycosylation sites, and potential Tyr sulfation sites within their N-terminal sequences (Fig. 1a and Supplementary Fig. 1). Thus, these sequences constitute a distinct evasin subclass and we have designated them "class A3 evasins".

Phylogenetic analysis revealed that class A3 evasins form a separate subclade of the previously defined class A1 evasins[13] (Fig. 1b and Supplementary Fig. 2). In particular, we identified class A3 evasin sequences in six species within the tick genus *Amblyomma* but not in any species from other tick genera. Thus, class A3 evasins constitute a specialised evasin family that has evolved after divergence of *Amblyomma* from other genera of ticks.

In addition to class A3 evasins, we also identified seven evasin-like sequences, all from tick species in the genus *Rhipicephalus*, that contain two additional cysteine residues relative to class A1 evasins, but in different positions from those of class A3 evasins (Supplementary Fig. 3). These sequences may represent a fourth subfamily of class A evasins.

### Class A3 evasin EVA-AAM1001 recognises multiple CC chemokines
To validate a class A3 evasin as a chemokine-binding protein, we expressed in *E. coli* and purified an evasin from *A. americanum* named EVA-AAM1001 (AAM for *A. Americanum* (Supplementary Fig. 4a); 10 for the ten Cys residues; 01 for the first member of this family; hereafter abbreviated to EVA-A; previously labelled with the codes AAM-02[16] or E1243[17]). The mass spectrum confirmed the presence of five disulfide bonds (Supplementary Fig. 4b), the distinct feature of this subclass of evasins. C-terminally Avi-tagged EVA-A was biotinylated, immobilised on a surface plasmon resonance (SPR) chip and screened for binding to all available human chemokines. EVA-A bound to 20 CC chemokines with affinities (equilibrium dissociation constant, $K_d$) of 250 nM or tighter (Fig. 1c, d; Supplementary Fig. 5, 11 and Supplementary Table 1), but did not bind to any CXC, $CX_3C$ or XC chemokines. EVA-A was previously found to bind many of the same chemokines, generally with similar affinities (Supplementary Fig. 6). Although statistical comparison of affinities is not possible because only a single $K_d$ value was reported[17], some apparent affinity differences could potentially be attributed to differences in expression systems, post-translational modifications, purity, biophysical methods and/or solution conditions for binding experiments. Whereas EVA-4 is also a broad-spectrum binder to CC chemokines[11], EVA-4 and EVA-A exhibit quite distinct chemokine affinity profiles (Supplementary Fig. 7). To verify that binding of class A3 evasins gives rise to functional inhibition of chemokines, we determined the effect of EVA-A on chemokine-stimulated activation of chemokine receptors. As expected, EVA-A inhibited the ability of CCL3, CCL5, CCL7, CCL8, CCL13, CCL14, CCL15 and CCL23 (but not CCL2) to stimulate receptor-mediated phosphorylation of extracellular signal-regulated kinase 1/2 (ERK) (Supplementary Fig 8). In addition, EVA-A blocked the chemokine-stimulated inhibition of cyclic AMP (cAMP) synthesis in a concentration-dependent manner (Fig. 1e). These data confirm that this class A3 evasin selectively binds and inhibits CC chemokines.

### Structure of EVA-A in complex with chemokines
To understand how class A3 evasins achieve chemokine binding and inhibition, we purified and crystallised complexes of EVA-A bound to human CCL7, CCL11, CCL16 and CCL17, and solved their structures at resolutions of 1.51–2.01 Å (Fig. 2a, b). All four structures display the same fold and 1:1 stoichiometry (Fig. 2b). The chemokines assume the expected compact globular tertiary structure composed of a three-stranded antiparallel β-sheet packed against an α-helix. The N-terminus and N-loop, which are critical for receptor activation, are linked to the secondary structure through the two conserved disulfide bonds. EVA-A displays a rigid core structure comprised of seven β-strands and one α-helix, stabilised and interconnected by five disulfide bonds. Consistent with the sequence comparisons, four are conserved with class A1 evasins, whereas the fifth disulfide, which connects the N-terminus to the fourth β-strand, is unique to class A3 evasins.

### Class A3 and class A1 evasins have subtle structural differences
Previously, structures have been reported for the class A1 evasin EVA-P974 (hereafter abbreviated to EVA-P) bound to CCL7 and CCL17[14]. Comparison of these to our structures of EVA-A bound to the same chemokines enabled us to identify similarities and distinctive aspects of chemokine recognition by class A3 evasins. The overall fold of EVA-A is similar to that of class A1 evasins, except that the α-helix has shifted

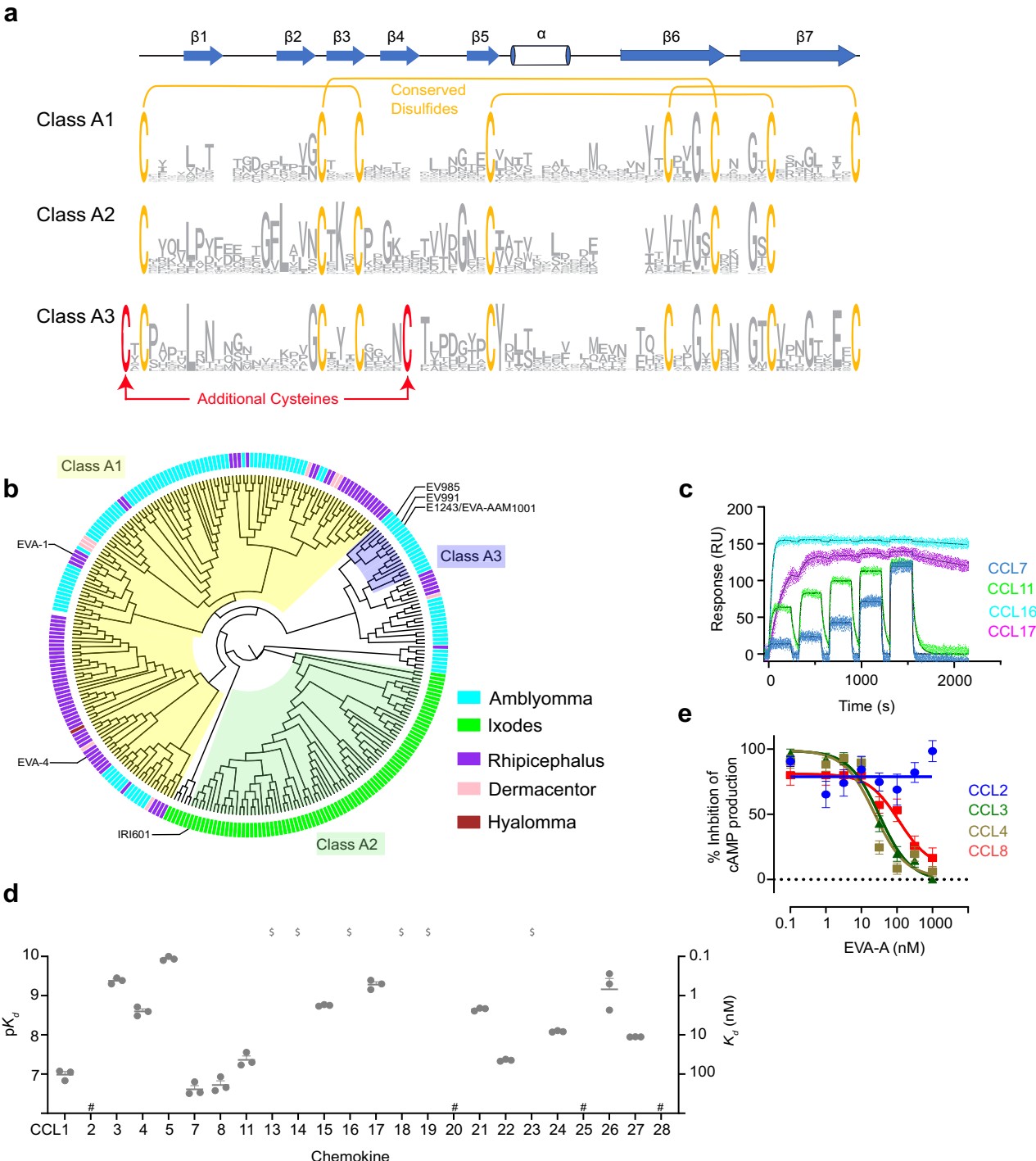

**Fig. 1 | Discovery and characterisation of class A3 evasins. a** Sequence logo based on alignments of class A evasins (class A1: $n = 149$, class A2: $n = 65$ and class A3: $n = 17$). Conserved cysteine and disulfide bond connectivity are indicated for class A1 evasins (orange). The additional cysteine and disulfide bond of class A3 are indicated in red. The secondary structure of EVA-P974 (class A1) is shown above the alignment. **b** A midpoint-rooted, neighbour-joining tree represented as a clado-gram, based on MUSCLE alignments of the class A evasins. Background colours show subclasses; A1 (light yellow), A2 (green) and A3 (purple). The genus of each node is indicated by coloured segments in the outer ring and shown in the legend. **c** Representative binding sensorgrams of human chemokines (5 injections at con-secutive concentrations of 31.25, 62.5, 125, 250 and 500 nM) measured by SPR using single-cycle kinetics. **d** Binding affinities ($K_d$) of EVA-A for CC chemokines, mea-sured by SPR. Data are presented as mean ± SEM from three independent

experiments. $, $K_d < 0.1$ nM; #, no measurable binding at 500 nM chemokine con-centration. **e** Concentration response curves showing the inhibition of chemokines CCL3, CCL4 and CCL8, but not CCL2, by EVA-A. FlpInCHO cells stably expressing CCR5 (for CCL3, CCL4 and CCL8) or CCR2 (for CCL2) and transfected with the cAMP biosensor CAMYEL, were treated with coelenterazine h (5 μM, 10 min), fol-lowed by forskolin (10 μM, 10 min) to induce cAMP production, followed by CCL3 (60 nM), CCL4 (80 nM), CCL8 (100 nM) or CCL2 (100 nM), either alone or pre-incubated with the indicated concentrations of EVA-A. cAMP was detected 10 min after chemokine addition. Data are represented as a percentage of the inhibition of cAMP production observed upon chemokine treatment in the absence of EVA-A, and presented as mean ± SEM from three or four independent experiments.

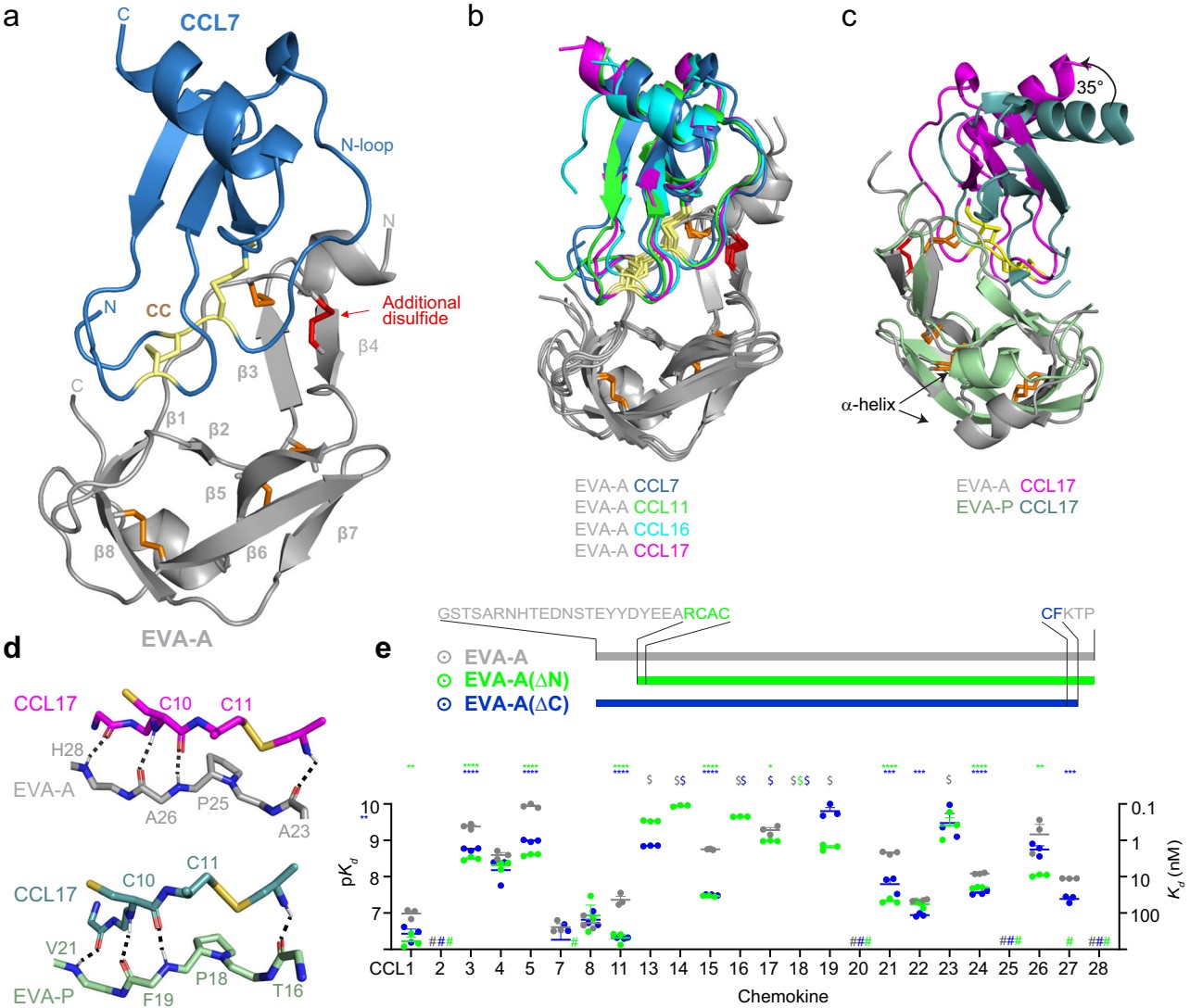

**Fig. 2 | Structure of EVA-A bound to the human CC chemokines. a** Overall structure of EVA-A (grey; conserved disulfides, orange; additional disulfide, red) in complex with CCL7 (sky blue; CC motif, yellow) showing all the major features as labelled. **b** Overlay of EVA-A complexes with CCL7 (blue), CCL11 (green), CCL16 (cyan) and CCL17 (magenta). **c** Cartoon representation of overlaid structures of EVA-A (grey) bound to CCL17 (magenta) and EVA-P (pale green) bound to CCL17 (deep teal) (PDB ID:7S4N) showing the different orientation of CCL17 by 35°.

**d** Conserved mode of CC chemokine recognition by EVA-A and EVA-P.
**e** Truncations of the N- and C-termini of EVA-A have subtle effects on the chemokine binding affinities of EVA-A. Data are presented as mean ± SEM from three independent experiments. $, $K_d < 0.1$ nM; #, no measurable binding at 500 nM chemokine concentration; *$p < 0.05$, **$p < 0.01$, ***$p < 0.001$, ****$p < 0.0001$, versus wild type EVA-A (one-way ANOVA with Šídák correction for multiple comparisons for each chemokine; or a two-tailed $t$ test if only a single comparison was possible).

by ~7–9 Å further from the chemokine binding site (Fig. 2c). Since this helix was already distal from the chemokine binding site, the shift is unlikely to influence chemokine recognition directly. Notably, recognition of the chemokine CC motif utilises the same four backbone-backbone hydrogen bonds in class A1 and A3 evasins (Fig. 2d). This mode of recognition is not compatible with insertion of additional residues between the two cysteine residues of the CC motif[14], (Supplementary Fig. 9) explaining why both subclasses are specific for CC over CXC or CX₃C chemokines. On the other hand, there is a consistent ~35° rotation of the chemokine between the available class A1 and class A3 evasin-bound structures (Fig. 2c), suggesting that class A3 evasins enable closer interactions with the N-loop region of the chemokine.

**The N- and C-termini of EVA-A are dispensable for chemokine binding**

In class A1 evasins, the evasin N-terminal region plays a critical role in chemokine N-loop recognition. Specifically, N-terminal truncation of

EVA-P, leaving only four residues before the first conserved Cys, completely abrogated binding to all 11 CC chemokine ligands[14]. In all four chemokine complexes described here, the N-terminus of EVA-A is in contact with the chemokine N-loop. Moreover, the EVA-A N-terminus has the potential to undergo post-translational tyrosine-sulfation when expressed in eukaryotic cells, thus enhancing chemokine affinity[18]. Nevertheless, we found that the N-terminus of EVA-A could be truncated, leaving just three residues before the first conserved Cys, with only small (albeit some significant) losses of affinity, allowing most chemokines to retain binding at affinities of ~0.1–100 nM (Fig. 2e; Supplementary Fig. 12 and Supplementary Table 2). This suggested, that EVA-A utilises alternative interactions from those of the N-terminal residues in EVA-P to enable high affinity binding to many CC chemokines.

One possible source of alternative binding affinity is the interactions between the EVA-A C-terminus and the N-terminus of the chemokines. The equivalent interactions make a significant contribution

to binding between EVA-P and some (but not all) cognate chemokines. However, again we found that truncation of the EVA-A C-terminus had minimal effects on chemokine affinity (Fig. 2e; Supplementary Fig. 13 and Supplementary Table 3).

### The fifth disulfide bond of class A3 evasins defines a "CC + 1" binding pocket

In all CC chemokines, the residue immediately following the CC motif (here designated the "CC + 1" residue) has substantial hydrophobic character (at least three hydrophobic carbon atoms) and the potential to form hydrophobic interactions; in some cases, this residue has been found to contribute to chemokine receptor binding and activation[19]. In the EVA-A:chemokine structures, the side chain of the chemokine CC + 1 residue (Tyr in CCL7; Phe in CCL11, Leu in CCL16 and CCL17) sits in a deep hydrophobic pocket within the evasin structure (Fig. 3a, b), possibly enabling the ~35° rotation of the chemokine relative to its position in class A1 evasin complexes. Notably, this "CC + 1 recognition pocket" is defined by several hydrophobic side chains and by the Cys22-Cys51 disulfide bond, the fifth disulfide that distinguishes class A3 from class A1 evasins (Fig. 3a expansion). Moreover, we found that mutation of Cys22 and Cys51 to Ser, giving EVA-A($C_8$), resulted in significantly or substantially diminished binding affinity (or loss of measurable binding) to most chemokines (Fig. 3c; Supplementary Fig. 14 and Supplementary Table 4). Nevertheless, this double mutant co-crystallised with CCL17, with the only structural changes being in the immediate vicinity of the mutated disulfide bond (Fig. 3d; Supplementary Fig. 10). Thus, it appears that the key function of the unique disulfide bond in class A3 evasins is to structurally define the CC + 1 recognition pocket.

### Aromatic "CC + 1" residues are disfavoured through negative selection

Considering that the CC + 1 recognition pocket is a major location of chemokine side chain interactions with EVA-A, we postulated that the interactions of this pocket would influence the binding selectivity of EVA-A amongst different chemokines. To verify the contribution of the CC + 1 residue to affinity, we mutated this residue to Ala in CCL7, CCL11 and CCL16. Surprisingly, CCL16(L19A) exhibited only slightly reduced affinity for EVA-A (Fig. 3e, f). Moreover, CCL7(Y13A) and CCL11(F11A) exhibited substantially increased affinity (~10,000 and ~200 fold, respectively) (Fig. 3e, f), suggesting that the interactions of the aromatic CC + 1 residues in these chemokines had imposed an energetic penalty that was relieved by mutation to Ala. The complex of EVA-A with CCL7(Y13A) is isostructural with the wild type CCL7 complex, except in the immediate vicinity of the mutation (Fig. 3g), indicating that the affinity increase can, indeed, be attributed to changes in side chain interactions. Thus, we conclude that the preference of EVA-A for CCL16 over CCL7 and CCL11 is because the aromatic residues in the latter two chemokines are energetically disfavoured, a phenomenon known as negative selection[20].

### Modifying chemokine selectivity by removal of negative selection

Considering the above finding that the interactions of aromatic CC + 1 residues of CCL7 and CCL11 with the hydrophobic pocket impose an energetic penalty against high affinity binding, we postulated that the affinity of EVA-A for these (and possibly other) chemokines could potentially be enhanced by alleviating this negative selection. To achieve this, we mutated residues Tyr44 and Leu39 in the CC + 1 binding pocket (Fig. 3a) to Ala and Pro, respectively; Pro was chosen as it occurs in the corresponding position to Leu39 in several class A3 evasins. We found that the Y44A and L39P mutants exhibited increased affinity to most chemokines with aromatic CC + 1 residues but decreased affinity to most chemokines with aliphatic CC + 1 residues (Fig. 4a; Supplementary Fig. 15, 16 and Supplementary Table 5, 6). In

the case of L39P, the affinity decreases were only small so this mutant bound to 21 of the 24 available CC chemokines with affinities tighter than 100 nM, compared to 17 chemokines for wild type EVA-A. Consistent with their increased affinity for chemokines with aromatic CC + 1 residues, both the Y44A and L39P mutants exhibited substantial enhancements in their potencies for inhibition of both CCL2 and CCL7 (Fig. 4b). In addition, both variants inhibited the receptor-mediated phosphorylation of ERK by CCL2 as well as by the same eight chemokines as wild type EVA-A (Supplementary Fig. 8).

To validate the structural basis of enhanced binding and inhibition of chemokines with aromatic CC + 1 residues, we determined the structures of EVA-A(Y44A) in complex with CCL2 and both mutants in complex with CCL7 (Fig. 4c). The structure of L39P showed the aromatic CC + 1 side chain (Tyr13) of CCL7 buried deeper within the CC + 1 binding pocket than would have been possible for wild type EVA-A, closely approaching the mutated residue (Pro39) and forming favourable π-stacking interactions with Tyr44 and His28 (Fig. 4c). The chemokine-bound structures of EVA-A(Y44A) are isostructural with the CCL7 complex of wild type EVA-A, but the mutation of Tyr44 to Ala avoids the repositioning of the bulky Tyr44 side chain, as required for binding of CCL7 to wild type EVA-A. Moreover, while wild type EVA-A favoured Ala mutants of aromatic CC + 1 residues over the corresponding wild type chemokines (Fig. 3f), the Y44A and L39P mutants either favoured the wild type chemokines or bound with comparable affinity to the wild type and mutant chemokines (Fig. 4d; Supplementary Fig. 17 and Supplementary Table 7, 8, 9). This indicated that the evasin mutants no longer display strong negative selection against binding to the aromatic CC + 1 residues of CCL7 and CCL11. Thus, the increased breadth of chemokine binding by EVA-A(L39P) resulted, at least in part, from the removal of negative selection.

### CC motif interactions are more rigid in class A3 than class A1 evasins

The above results indicate that interactions outside the CC motif recognition region (the N- and C-termini and the CC + 1 recognition pocket) make only minor contributions to the overall binding energy of EVA-A for chemokines. In contrast, we showed previously that the N-terminus is a major source of chemokine binding free energy for the class A1 evasin EVA-P. Nevertheless, EVA-A binds to chemokines with comparable or higher affinity than EVA-P. Consequently, it appears that the CC motif recognition region contributes more free energy to binding in the case of EVA-A.

The backbone-backbone hydrogen bonds involved in CC motif recognition are isostructural in chemokine complexes of EVA-A and EVA-P (Fig. 2c). Therefore, we considered whether there may be differences in dynamics at this interface or in the unbound evasins, which we investigated using molecular dynamics (MD) simulations of both the free evasins (N-terminally truncated) and their CCL17 complexes. Throughout the MD trajectories, the evasin structures remained very similar to the corresponding crystal structures (Supplementary Fig. 18). Interestingly, despite the additional disulfide bond, the structure of unbound EVA-A displayed higher flexibility (backbone root-mean-square fluctuations, RMSF) than EVA-P, including in the CC motif recognition region (β1-strand) (Fig. 5a). However, upon binding to CCL17, EVA-A became more rigid (lower root-mean-square fluctuation (RMSF) values) than EVA-P (Fig. 5b). Moreover, the inter-chain positions between CCL17 and EVA-A exhibited less variability than that between CCL17 and EVA-P (Fig. 5c), suggesting a more stable complex formed between CCL17 and EVA-A. This is consistent with EVA-A forming high-occupancy hydrogen bonds with CCL17 residues Cys50 and Asp33, thus "clamping" each end of the CC motif region (Fig. 5d, Supplementary Table 10). In contrast, EVA-P forms a single high-occupancy hydrogen bond in the centre of this region (Fig. 5e, Supplementary Table 10). Taken together, these MD findings suggest that the CC motif of CCL17 forms more stable interactions with the class A3

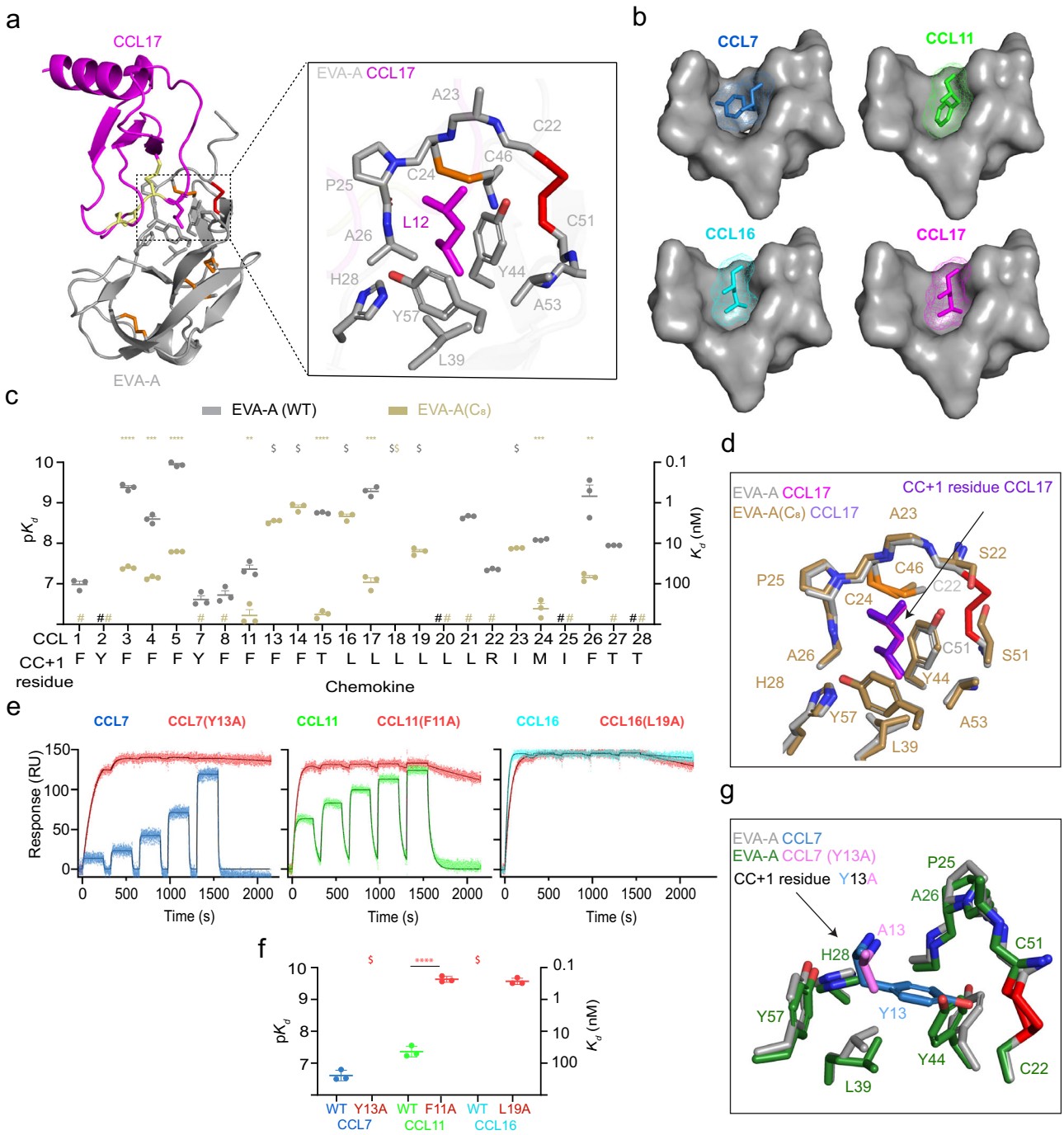

**Fig. 3 | The fifth disulfide defines a critical binding pocket for chemokine target selectivity (CC + 1 residue binding). a** Cartoon representation of EVA-A CCL7, with expanded view showing the EVA-A residues (grey, additional disulfide: red) of the hydrophobic pocket holding the CC + 1 residue (magenta) of CCL17. **b** Surface representations of the EVA-A binding pocket (grey) fitting the CC + 1 residue of four different chemokines; CCL7 (sky blue), CCL11 (green), CCL16 (cyan) and CCL17 (magenta). **c** Chemokine affinities ($K_d$) of EVA-A (grey) and EVA-A($C_8$) (sand). Data are presented as mean ± SEM from three independent experiments. \$, $K_d < 0.1$ nM; #, no measurable binding at 500 nM chemokine concentration; *$p < 0.05$, **$p < 0.01$, ***$p < 0.001$, ****$p < 0.0001$, versus wild type EVA-A (two-tailed $t$ test with Holm-Šídák correction for multiple comparisons). **d** Overlay of the binding pocket

residues of EVA-A (grey) bound to the CC + 1 residue of CCL17 (magenta) and EVA-A($C_8$) (sand) bound to the CC + 1 residue of CCL17 (purple). **e** Representative SPR sensorgrams for binding of EVA-A to wild type chemokines (CCL7, sky blue; CCL11, green; and CCL16, cyan) and CC + 1 residue-mutated versions of each chemokine (red) measured using single-cycle kinetics (5 chemokine injections at consecutive concentrations of 31.25, 63.5, 125, 250 and 500 nM). **f** EVA-A affinities ($K_d$) for wild type and CC + 1 residue-mutated chemokines (coloured as in **e**). Data are presented as mean ± SEM from three independent experiments. \$, $K_d < 0.1$ nM; ****$p < 0.0001$, versus wild type chemokine (two-tailed $t$ test). **g** Overlay of the binding pocket residues of EVA-A (grey) bound to CCL7 (sky blue) and EVA-A (forest green) bound to CCL7(Y13A) (violet).

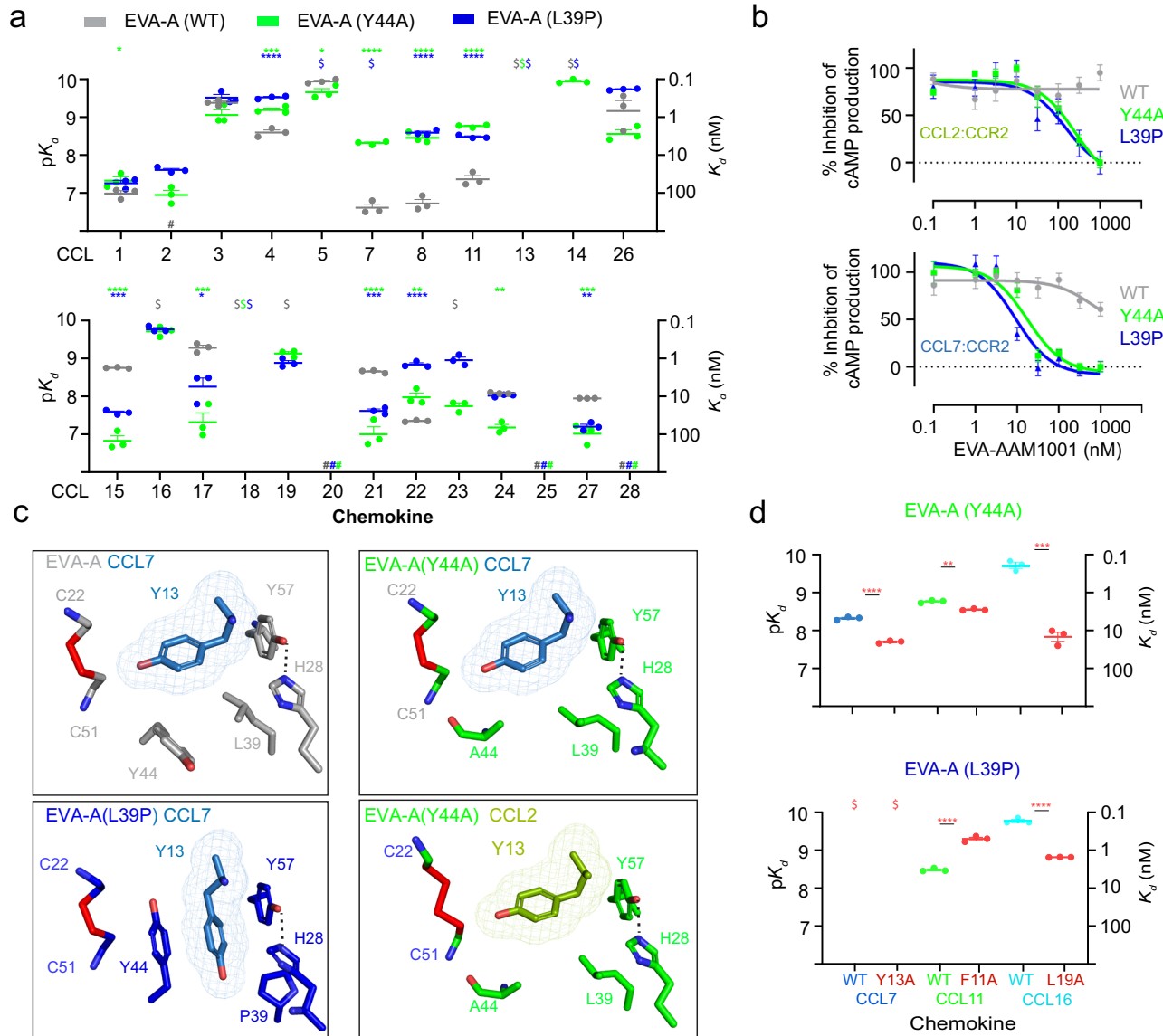

**Fig. 4 | Engineering of the EVA-A hydrophobic pocket. a** CC chemokine affinities ($K_d$) of EVA-A (grey), EVA-A(Y44A) (green) and EVA-A(L39P) (blue). Upper panel, chemokines with aromatic CC + 1 residues; lower panel, chemokines with aliphatic CC + 1 residues. Data are presented as mean ± SEM from three independent experiments. $, $K_d < 0.1$ nM; #, no measurable binding at 500 nM chemokine concentration; *$p < 0.05$, ****$p < 0.0001$, versus wild type EVA-A (one-way ANOVA with Šídák correction for multiple comparisons; or a two-tailed *t* test if only a single comparison was possible). **b** Concentration-response curves showing the inhibition of CCL2 (100 nM) (top Panel) and CCL7 (100 nM) (bottom panel) by EVA-A, EVA-A(Y44A) and EVA-A(L39P). FlpInCHO cells stably expressing CCR2 transfected with the cAMP biosensor CAMYEL, were treated with coelenterazine h (5 μM, 10 min), followed by CCL2 (100 nM) or CCL7 (100 nM), either alone or pre-incubated with the indicated concentrations of EVA-A, followed by forskolin (10 μM, 10 min) to induce cAMP production. cAMP was detected 10 min after chemokine addition. Data are represented as a percentage of the inhibition of cAMP production observed upon chemokine treatment in the absence of EVA-A, and presented as mean ± SEM from three independent experiments. **c** Interactions of chemokine CC + 1 residues (sticks with mesh) with hydrophobic pocket side chains (sticks) of EVA-A, EVA-A(Y44A) and EVA-A(L39P). Top (left to right): EVA-A (grey) bound to CCL7 (sky blue) and EVA-A(Y44A) (green) bound to CCL7 (sky blue). Bottom (left to right): EVA-A(L39P) (deep blue) bound to CCL7 (sky blue) and EVA-A(Y44A) (green) bound to CCL2 (olive). **d** EVA-A(Y44A) (top) and EVA-A(L39P) (bottom) affinities ($K_d$) for wild type and CC + 1 residue-mutated chemokines (coloured as in Fig. 3e, f). Data are presented as mean ± SEM from three independent experiments. $, $K_d < 0.1$ nM; **$p < 0.01$, ***$p < 0.001$, ****$p < 0.0001$, versus wild type chemokine (two-tailed *t* test).

evasin EVA-A than with the class A1 evasin EVA-P. While the reduced flexibility of EVA-A suggests a greater entropic cost to forming these interactions (compared to EVA-P), clearly the enthalpic advantage of forming the interactions outweighs this cost, otherwise they would not form.

## Discussion

The migration, activation, differentiation and survival of leukocytes in inflammation is regulated through the activation of their chemokine receptors by numerous chemokines expressed in a wide variety of tissues[21]. Whereas pharmacological targeting of chemokine receptors in anti-inflammatory therapy has met with limited success, ticks achieve effective suppression of inflammation by secreting a cocktail of evasin proteins with different chemokine target selectivities. However, the goal of using such evasins as anti-inflammatory therapeutics will require a more nuanced approach, most likely requiring evasin engineering to achieve selectivity towards a chosen chemokine subset. To this end, it is necessary to identify natural evasins with diverse chemokine-binding properties and to understand the molecular basis of their chemokine selectivity. In this study, we identified and

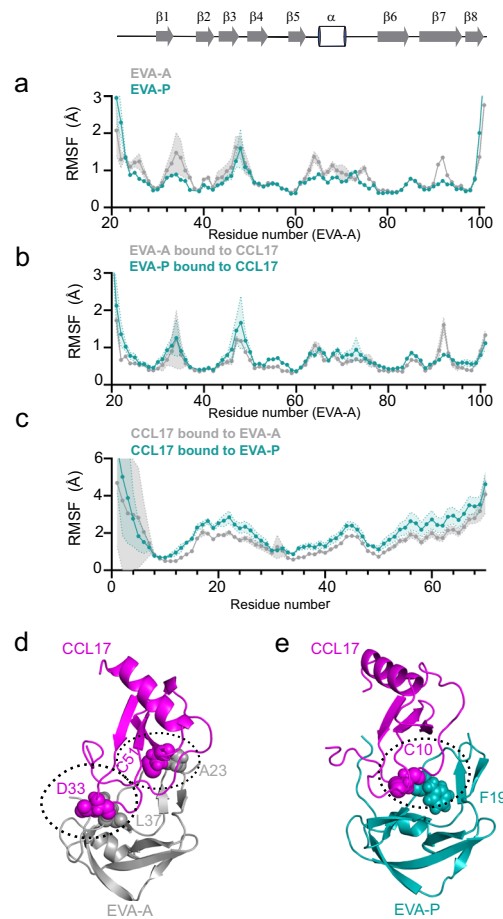

**Fig. 5 | Class A3 evasins achieve broad chemokine binding through flexible structure. a, b** Plots comparing the root-mean-square fluctuation (RMSF) values during MD simulations for EVA-A (grey) and EVA-P (green teal). The secondary structure of EVA-A in the crystal structure is shown at the top. **a** Free evasins only. **b** Bound state with chemokine. **c** RMSF of CCL17 bound to EVA-A (grey) and EVA-P (teal) with MD trajectories aligned at the evasins. Data are presented as mean ± SEM from three independent simulations. **d** The EVA-A:CCL17 complex forms high occupancy intermolecular hydrogen bonds at both ends of CC the motif. **e** The EVA-P:CCL17 complex forms a single high occupancy intermolecular hydrogen bond in the centre of the interface.

characterised class A3 evasins, showing how their structural and dynamical features allow recognition of numerous CC chemokines.

Comparison of class A3 evasins to class A1 and A2 evasins provides insight into the development of anti-inflammatory, chemokine-binding proteins during tick evolution. Phylogenetic analysis of evasin sequences suggested that class A1 and A2 evasins may have evolved from a common ancestor before the divergence of prostriate and metastriate lineages (up to ~250 MYA) but that class A2 evasins appeared before prostriate tick diversification (~217 MYA) and class A1 evasins appeared before metastriate tick diversification (~138 MYA)[13]. By contrast, the species encoding class A3 evasins – *A. triste, A. parvum, A. cajennense* (Cayenne tick), *A. americanum* (lone star tick), *A. aureolatum*, and *A. tuberculatum* (gopher tortoise tick) – are all native to the Americas. Based on their restricted species distribution, we propose that class A3 evasins represent a specialised subfamily that evolved from class A1 evasins after the divergence of *Amblyomma* from other metastriate lineages (~60–120 Mya)[13]. It remains to be determined whether *Amblyomma* species outside the Americas also encode class A3 evasins.

The emergence of class A3 evasins in American *Amblyomma* ticks suggests that class A3 evasins confer a competitive advantage for these species. Considering that *Amblyomma* ticks infest a wide variety of species[22], it is possible that the broad chemokine binding profiles of class A3 evasins facilitate the spread of ticks amongst different host species. On the other hand, ticks from other genera, such as *Rhipicephalus*, also infest a variety of species[23–25], so the occurrence of class A3 evasins in *Amblyomma* species may not be a dominant factor in controlling host compatibility. Indeed, an alternative strategy to achieve broad chemokine inhibition is simply to secrete a cocktail of evasins with different chemokine selectivities, as discussed previously for the species *R. pulchellus*[11].

Evasins have potential as chemokine-targeted anti-inflammatory therapeutics and as tools for research. However, for optimal utility, they will need to be engineered to have the appropriate target selectivity for any given application. The data presented here suggest that class A3 evasins differ from class A1 evasins not only in their sequences but also in several key aspects of their chemokine recognition. First, the N- and C-terminal regions of EVA-A (and perhaps other class A3 evasins) play relatively minor roles in chemokine recognition. Second, the additional disulfide bond of class A3 evasins enables the formation of a deep hydrophobic pocket for binding to the chemokine CC+1 residue. Third, the structures of class A3 evasins appear to be more flexible than class A1 evasins. All these features have implications for the selectivity of chemokine binding by class A3 evasins.

In all CC chemokines, the CC+1 residue is large, has hydrophobic character, and protrudes from the protein surface, playing an important role in chemokine receptor binding and activation[19]. EVA-A accommodates this residue within a hydrophobic pocket, which is formed by the disulfide bond that defines class A3 evasins as well as the side chains of residues Leu39, Tyr44 and Tyr57. These residues are highly conserved across the 14 class A3 evasin sequences (Supplementary Fig. 19), indicating that the CC+1 binding pocket is probably a conserved feature of class A3 evasins. The detailed structure of the hydrophobic pocket may enable a class A3 evasin to exhibit preferences for chemokines with different CC+1 residues, as observed for wild type EVA-A. Alternatively, the pocket may accommodate a relatively wide variety of CC+1 residues, as observed here for the EVA-A(L39P) mutant. It is noteworthy that five of the 14 identified class A3 evasins have proline at the equivalent position of Leu39 in EVA-A (Supplementary Fig. 19), so the wild type forms of these evasins may recognise a fairly broad spectrum of CC chemokines.

In theory, selective binding may be accomplished by the formation of energetically favourable interactions with the preferred target(s) (positive selection) or the formation of energetically unfavourable interactions with the alternative target(s) (negative selection), or a combination of the two strategies. Here, we found that the preference of EVA-A for chemokines with aliphatic CC+1 residues results from negative selection against some chemokines with aromatic CC+1 residues (Fig. 3e, f). Negative selection has also been found to regulate selective binding in other biomolecular systems. For example, *Drosophila melanogaster* neuronal recognition proteins, Dpr (Defective proboscis extension response) and DIP (Dpr interacting proteins) achieve binding specificity via negative constraints. These negative constraints include: Coulombic repulsion, burial of an unsatisfied charged group in a pocket, steric clashes due to larger amino acids in a hydrophobic pocket, and smaller amino acids forming cavities in a pocket diminishing the binding affinity[26]. Similarly, negative selection has been suggested to influence the specificity of yeast Src homology 3 (SH3) domains for cellular signalling partners[27]. The observation of this molecular selectivity strategy across such diverse protein families suggests that it may be utilised widely in biomolecular recognition.

Identifying the structural origin of negative energy contributions at the CC+1 binding pocket enabled us to reprogram EVA-A as a more promiscuous CC chemokine binder. Conversely, negative design elements can be incorporated at binding interfaces to engineer

protein-protein complexes with improved selectivity[28,29]. Such negative elements could include hydrogen bonds, charge interactions or hydrophobic interactions with mismatched complementarity to undesired binding partners[30]. Indeed, incorporation of such elements into EVA-A(L39P) may facilitate tailoring of its selectivity to different subsets of CC chemokines, as needed in different therapeutic contexts.

Considering that the EVA-A termini and CC + 1 pocket contribute little to binding affinity (or even oppose binding), most of the favourable binding energy must be attributed to the interactions of the EVA-A β1-strand with the chemokine CC motif. The observation that EVA-A is more flexible in the free state but more rigid in the chemokine-bound state, in comparison to EVA-P, is consistent with EVA-A forming tighter interactions. This is further supported by the lower flexibility of CCL17 and the increased frequency of high-occupancy hydrogen bonds for the class A3 evasin:chemokine complex. Indeed, the flexibility of unbound class A3 evasins may facilitate high-affinity binding to many chemokines by enabling induced fit of the evasin and chemokine surfaces to optimise binding energy. Thus, we propose that the structural and dynamical properties of class A3 evasins have evolved from class A1 evasins to optimise binding to a broad range of CC chemokines, within a single species or across the numerous host species of *Amblyomma* ticks.

In summary, we have identified a subclass (class A3) of tick evasins, which have evolved from class A1 evasins but have broader chemokine-binding potential than most other evasins identified to date. We propose that class A3 evasins serve as an excellent platform for engineering proteins with desired chemokine binding selectivity. Such proteins have a variety of potential applications as diagnostic and research reagents or as therapeutic anti-inflammatory agents.

## Methods

### Phylogenetic analysis
A phylogenetic tree was generated based on a previously published set of evasin sequences[13]. Protein sequences were aligned using MUSCLE[31] and a midpoint rooted neighbour-joining tree was generated (R version 4.1.0, ape 5.6)[32] and visualized as a cladogram (ggtree 3.2.1)[33]. Complete sequences from clades containing *Ambylomma* class A3 evasins, class A2 evasins and class A1 evasins were aligned, trimmed to the cysteine motif region, with low frequency columns representing rare insertions removed for clarity, and visualized as a sequence logo (ggseqlogo 0.1)[34].

### Plasmid constructs
The DNA sequences encoding EVA-A, CCL2, CCL7, CCL11, CCL16 and CCL17 were purchased as gBlocks from Integrated DNA Technologies or Genscript. The sequences were cloned into the vector pET-28a (Novagen) encoding an N-terminal His$_6$ tag and a Small Ubiquitin-like Modifier (SUMO) tag. All mutants of EVA-A and chemokines were generated by golden gate assembly using BsaI enzyme. All constructs used in this study were confirmed by DNA sequencing.

### Protein purification
Plasmids with the gene of interest were transformed into *E. coli* (Rosetta-gami 2 (DE3)) competent cells and cultured in 2YT medium (supplemented with 30 μg/ml kanamycin) at 37 °C for protein expression. The protein was induced at the optical density at 600 nM (OD$_{600}$) 0.6-0.7 by adding 0.5 mM isopropyl β-D-1-thiogalactopyranoside (IPTG) to the culture medium. After overnight culture at 20 °C, cells were harvested by centrifugation (5,422 g, 10 min, 4 °C) and resuspended with buffer A (20 mM Tris pH 8.0, 500 mM NaCl, 10% glycerol). Cells were disrupted by sonication. After centrifugation (29,097 g, 40 min, 4 °C), the clarified cell lysate was loaded into a 5 ml His-Trap FF nickel column (Cytiva) and the SUMO fused protein attached to the column was eluted with buffer A containing 500 mM of imidazole. To remove imidazole, the protein solution was dialysed overnight in buffer A and, the next day dialysed protein was incubated with ULP1 protease (60 min, 25 °C) to cleave His$_6$-SUMO from the fusion protein. After complete cleavage, the protein solution was passed through the nickel column to remove the His$_6$-SUMO tag. The flow-through was concentrated and purified on a Superdex 75 size exclusion chromatography column (Cytiva) in a buffer 10 mM HEPES pH 7.5, 150 mM NaCl, 5% glycerol. Fractions containing protein of the expected size were pooled and injected onto a 214TP C4 (250 × 10 mm, 5 μm) reversed-phase high-performance liquid chromatography (RP-HPLC) column (Vydac), attached to an Agilent Technologies 1200 series instrument, pre-equilibrated with water containing 0.001% trifluoracetic acid, and eluted using a 1% per min gradient of acetonitrile containing 0.001% trifluoracetic acid. The pure and homogenous protein eluted in the range ~30–45% of acetonitrile. Pooled fractions were lyophilised and its quality assessed before subsequent experiments. The final yield of protein was ~0.5–0.8 mg per L of culture.

### Mass spectrometry
Samples were analysed by tandem liquid chromatography-mass spectrometry (LC-MS) using a quadrupole TOF mass spectrometer (MicroTOFq, Bruker Daltonics, Bremen, Germany) coupled online with a 1200 series nano HPLC (Agilent Technologies, Santa Clara, CA, USA). Samples were injected onto a Zorbax 300SB reversed phase column with 95% buffer A (0.1% formic acid) at a flow rate of 300 nl/min. The proteins were eluted over a 30-min gradient to 70% B (80% acetonitrile 0.1% formic acid). The eluant was nebulised and ionised using a Bruker nanoESI source electrospray needle with a capillary voltage of 4500 V dry gas at 180 °C, a flow rate of 50 μl/min and nebuliser gas pressure at 300 mbar. Prior to analysis, the qTOF mass spectrometer was calibrated using 1:50 dilution tuning mix (Agilent technologies, Santa Clara, CA, USA). The spectra were extracted and deconvoluted using Data explorer software version 3.4 build 192 (Bruker Daltonics, Bremen, Germany).

### Surface plasmon resonance
For binding experiments, EVA-A and its variants were expressed with a C-terminal linker (GGGGS)$_3$ and AVI tag (GLNDIFEAQKIEWHE) and purified as described above. Before any binding experiments, the protein was reconstituted in a biotinylation buffer (10 mM Tris pH 8.0) and biotinylated using BirA protease in the presence of 500 mM Bicine pH 8.3, 500 μM D-biotin, 100 mM magnesium acetate and 100 mM ATP, then purified by size exclusion chromatography. Chemokine binding affinity to EVA-A and its variants was assessed using SPR on a Biacore T100 (Cytiva) instrument, using a Biotin CAPture kit Series S (Cytvia) chip and running buffer 10 mM HEPES pH 7.5, 500 mM NaCl, 0.002% Tween 20, 3 mM EDTA, 1 mg/ml carboxymethyl dextran. All binding experiments were performed at 25 °C. AVI-tagged EVA-A variants were captured on a streptavidin coated CAPture chip by flowing the evasin (~0.1 to 0.2 μM) for 60 s at the flow rate of 10 μl/min to obtain a signal of 200–250 response units (RU). Initially, chemokines were screened for EVA-A using single-cycle kinetics, injecting 500 nM of chemokines for 240 s at 30 μl/min, followed by a dissociation time of 600 s. To determine binding affinities, chemokines were injected at five concentration (31.25 nM to 500 nM; two-fold serial dilutions) using single-cycle kinetics (30 μl/min, 180 s per injection). The surface was regenerated using 6 M guanidine hydrochloride and 0.3 M NaOH for 120 s at 30 μl/min. The association rate constants ($k_a$), dissociation rate constants ($k_d$) and binding affinities (p$K_d$ = −log($K_d$)) were determined by fitting the sensorgrams with 1:1 binding kinetics using Biacore T100 Evaluation Software 2.0.4. Each experiment was repeated three times independently.

### Crystallisation of EVA-A chemokine complexes
The complexes of EVA-A with different chemokines were made by mixing in a 1:1.1 ratio and purified using size exclusion

chromatography. The eluted protein complexes were concentrated to ~20 mg/ml and subjected to crystallisation trials in commercial screens: JCSG-plus (Molecular Dimensions), Wizard (Molecular Dimensions), Crystal 20 (HAMPTON RESEARCH) and Index (HAMPTON RESEARCH) at the Monash Macromolecular Crystallisation Facility (MMCF). The CrystalMation system (Rigaku) with a Mosquito drop dispenser was used to produce drop volumes of 100 nl and a mother liquor reservoir volume of 50 μl in a 96-well format. Initial crystal hits of EVA-A CCL7 (0.1 M NaC$_2$H$_3$O$_2$, 2 M (NH4)$_2$SO$_4$ pH 4.5), EVA-A CCL11 (0.1 M Bris Tris pH 5.4, 2 M (NH$_4$)$_2$SO$_4$), EVA-A CCL16 (1.26 M (NH$_4$)$_2$SO$_4$, 0.1 M HEPES pH 7.5) and EVA-A CCL17 (30% W/W PEG 8 K, 0.1 M NaC$_2$H3O$_2$, 0.2 M LiSO$_4$ pH 4.5) obtained in coarse screening were optimised in hanging drops and other mutant crystals were obtained in their native crystallisation conditions. Crystal diffraction data was collected at MX1 and MX2 beamlines at the Australian Synchrotron. The diffraction data sets were indexed and processed using XDS[35] and scaled with AIMLESS[36] using the CCP4 program suite[37]. Data collection and refinement statistics are summarised in Supplementary Table 11.

### Structure solution and refinement

The EVA-A CCL17 structure was solved using the Auto-rickshaw pipeline[38]. For all other structures, EVA-A CCL17 was used as a search model using phaser to solve the structures. BUCCANEER[39], PHENIX[40] and COOT[41] were used to build and improve the initial structures. All refinement statistics for all the structures are given in Supplementary Table 11. Figures were generated using PyMOL 2.5.1.

### cAMP inhibition assay

cAMP-based bioluminescence resonance energy transfer (BRET) biosensor assay was used to determine the potency of EVA-A and its variants to inhibit the chemokine signalling through their cognate chemokine receptors[42]. For this purpose, FlpInCHO cells stably expressing the chemokine receptors CCR2 or CCR5 grown overnight in 10 cm dishes using Dulbecco's Modified Eagle Medium (DMEM) + GlutaMAX (Gibco, ThermoFisher Scientific) supplemented with 5% fetal bovine serum (FBS) and 1% penicillin/streptomycin (P/S) were transiently transfected with the CAMYEL biosensor using polyethylenimine (PEI) (Polysciences, Inc.) at a DNA: PEI ratio of 1:6 (w/w). The transfection mixture containing 5 μg of CAMYEL DNA, 30 μg of PEI in 500 μl of PBS was thoroughly vortexed and incubated at room temperature before adding the mixture to the cells (10 cm culture dish containing four million adherent cells). After 24 h of transfection, the next day cells were detached from a 10 cm culture dish using PBS-EDTA. Detached cells were seeded (50,000 cells/well) in a white 96-well plate (culture plates; PerkinElmer) and incubated overnight at 37 °C, 5% CO2. Cells were washed twice and equilibrated with Hank's Balanced Salt Solution (HBSS) for 30 min at 37 °C followed by the addition of the Rluc substrate coelenterazine-h (Nanolight Technology) at a final concentration of 5 μM for 10 min[43]. For initial experiments, concentration-response curves of chemokines were measured by activating the chemokine receptors with different concentrations of chemokines for 10 min, after which forskolin (final concentration 10 μM, Sigma Aldrich) was added and the cells incubated for 10 min before the final read. The Rluc and yellow fluorescent protein emissions were then measured at 475 and 535 nM, respectively, using a BRET one plus module in a BMG Labtech PHERAstar FS plate reader and data were extracted using MARS 3.32. Subsequently, evasin-mediated inhibition was determined by incubating cells with chemokines (EC80) alone or pre-incubated with the different concentrations of EVA- A variants, and the BRET ratio was measured. Data are presented as the BRET ratio, the percentage inhibition of forskolin-induced cAMP production by chemokines.

### ERK phosphorylation assay

Activation of chemokine receptors typically results in phosphorylation of extracellular signal-related kinase (ERK). Here, ERK phosphorylation was quantified using the AlphaLISA SureFire Ultra p-ERK1/2 (Thr202/Tyr204) assay (PerkinElmer). Human monocytic leukemia (THP-1) cells that endogenously express chemokine receptors (CCR1 and CCR2) were cultured in RPMI 1640 GlutaMAX™ (Gibco) supplemented with 10% FBS and 1% penicillin/streptomycin. The cells were harvested into 96-well culture plates at a density of 100,000 cells per well in an assay buffer consisting of HBSS containing 0.1% bovine serum albumin, 1 mM Ca$^{2+}$, and 1 mM Mg$^{2+}$. The cells were then incubated at 37 °C for 2 hours in serum-free medium for serum starvation, then stimulated for 5 min with either chemokine alone or chemokine-evasin mixtures. All chemokines and evasins were diluted in assay buffer, and each chemokine was mixed with each evasin (wild type or mutant) and incubated for 20 min at room temperature prior to cell stimulation. The stimulated cells were then lysed using SureFire Ultra-lysis buffer. After 10 min, 10 μl of the cell lysate from each well was transferred to a single well in a 384-well plate and allowed to incubate with 5 μl acceptor beads for 1 hour, followed by the addition of 5 μl of donor beads for an additional 1 hour in the dark. Phosphorylated ERK (pERK) was quantified from the luminescence signal detected by a plate reader (BMG Labtech Pherastar) using the AlphaScreen module with excitation and emission wavelengths of 680 and 615 nm, respectively. All cell treatments were tested in duplicate in four independent experiments. Data for each chemokine:evasin mixture were normalised to the maximum pERK signal for the chemokine alone. Statistical analysis was carried out by one-way ANOVA with Dunnett's multiple comparisons test.

### MD simulations

MD simulations for free N-terminal truncated evasins and in complex with CCL17 were conducted using Desmond[44] (Schrodinger Release 2022-3) and trajectories were visualized and analysed in VMD 1.9.4[45]. The N-terminally truncated EVA-P contains residues 13-97 and N-terminally truncated EVA-A contains residues 21-101. For MD simulations of the free evasins, two starting conformations were extracted for EVA-P or EVA-A from the crystal structures in complex with CCL7 and CCL17. For MD simulations of N-terminal truncated EVA-P or EVA-A in complex with CCL17, the starting conformations were extracted from the corresponding crystal structures.

The crystal structures were first prepared using Protein Preparation Wizard[46] (Schrodinger Release 2022-3) during which hydrogen atoms were added, all crystallographic water molecules were removed, and H-bond assignments were optimized using PROPKA (Schrodinger Release 2022-3) for pH 7.4. The structures were then minimized, converging heavy atoms to RMSD value of 0.3 Å. Each prepared structure was then placed at the centre of an orthorhombic box filled with explicit TIP3P water molecules, with a buffer of 10 Å to the edge of the box in all three directions. The protein chain was rotated to minimize the box volume. The system was then neutralized by adding Na$^+$ or Cl$^-$ atoms. A table describing all the MD system setups is provided in Supplementary Table 12.

For free N-terminal truncated evasins, a total of three replicate MD simulations, (1 μs each, 3 μs total), were conducted for EVA-P and EVA-A. Two replicate MD simulations were conducted starting from the EVA-P/ EVA-A conformation from the CCL7 complex and one additional MD simulation was conducted from the EVA-P/ EVA-A conformation from the CCL17 complex. For evasin-CCL17 complexes, three replicate MD simulations (1.5 μs each, 4.5 μs total) were conducted for EVA-P and EVA-A. The MD simulations were conducted with OPLS4 forcefield, at a constant temperature and pressure (310 K and 1 atm) using Langevin thermostat and barostat. The system was relaxed before simulation. Coulomb interactions were cut off beyond 9 Å. MD trajectory was written out every 1 ns.

Trajectories from the equilibrated time periods of the simulations were used for analysis, which correspond to total of 2.4 µs for free EVA-P (Supplementary Fig. 20a), 2.7 µs for free EVA-A (Supplementary Fig. 20b), 3 µs for EVA-P in complex with CCL17 (Supplementary Fig. 20b, c), and 3 µs for EVA-A in complex with CCL17 (Supplementary Fig. 20d, e).

## Quantification and statistical analysis

All graphical and statistical analyses were performed using GraphPad Prism 9.0.1. Data are representative of at least three independent experiments and presented as mean ± standard error of the mean (SEM). The number of repeats for each experiment and detailed descriptions of statistical tests are specified in the results section and/or the respective figure legends.

## Reporting summary

Further information on research design is available in the Nature Portfolio Reporting Summary linked to this article.

## Data availability

The coordinates and structure factors have been deposited in the Protein Data Bank under the following accession codes: EVA-A:CCL7, 7SCU; EVA-A:CCL11, 7SCS; EVA-A:CCL16, 7SCT; EVA-A:CCL17, 7SCV; EVA-A(Y44A):CCL2, 8FJ0; EVA-A(Y44A):CCL7, 8FK6; EVA-A(L39P):CCL7, 8FK8, 8FK8, EVA-A(C₈):CCL17, 8FJ2; and EVA-A:CCL7(Y13A), 8FJ3.

MD simulations files (initial and final coordinates) are publicly available at https://doi.org/10.5281/zenodo.8106592. The data that support the findings of this study are available within the paper and its Supplementary Information files. Source data are provided with this paper.

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

## Acknowledgements

We thank David Steer (staff of Monash Proteomics & Metabolomics Facility) for assistance with mass spectroscopy and Geoffrey Kong (staff of Monash Macromolecular Crystallisation Facility) for assistance with Nano-DSF and setting up crystallisation plates. This research was sup-ported by National Health and Medical Research Council Project Grants APP1140867 (M.J.S., R.J.P. and M.C.J.W.) and APP1140874 (M.J.S.) and Investigator Grant APP1174941 (R.J.P.), a Bridging Postdoctoral Fellow-ship from Monash University (R.P.B.), a Faculty Platform Access Grant from Monash University (R.P.B. and M.J.S.), and an Early Career Research Support Fund award from Monash University (R.P.B.). This research was undertaken in part using the MX2 beamline at the Australian Synchro-tron, part of ANSTO, and made use of the Australian Cancer Research Foundation (ACRF) detector. MD simulations were performed on the Rāpoi High-Performance Computing facility of Victoria University of Wellington.

## Author contributions

S.R.D., P.A., R.P.B., R.J.P. and M.J.S. conceived and designed the experiments; S.R.D., P.A., R.P., W.J., A.P., S.P. and R.P.B. carried out the experiments; S.R.D., P.A., W.J., A.P., M.C.J.W., R.P.B. and M.J.S. analysed the data; R.P.B. and M.J.S. supervised the studies; and S.R.D., P.A., R.P.B. and M.J.S. wrote the paper with input from all authors.

## Competing interests

The authors declare no competing interests.
