## [Peer review file · Nature Communications]

REVIEWER COMMENTS

Reviewer #1 (Remarks to the Author):

Evasins are naturally occurring leads for novel chemokine-neutralizing therapeutics and are thus of broad relevance to inflammatory disease. In this manuscript the authors elucidated the structure of a biochemically well characterised class A evasin "EVA-A", (a.k.a. AAM01/E1243_AMBAM/P1243) which is known to bind over 20 chemokines, in complex with chemokines CCL7, CCL11, CCL16, and CCL17. They show that EVA-A has a novel feature hitherto not observed in class A evasins i.e., the presence of 5 rather than 4 disulfide bridges. They show that the 5th disulfide bridge forms a hydrophobic pocket that binds the CC+1 residue. They mutate the Cys residues forming this disulfide bridge to explore its role in binding. They use alanine-mutagenesis of the CC+1 residue in the chemokine to explore the importance of this residue and show that for CCL7 and CCL11 such mutation enhances affinity. By mutating Y44 and L39 in the EVA-A-CC+1 binding pocket, they show that there was increased binding affinity and inhibitory potency against chemokines CCL2 and CCL7. In summary, the authors identify a novel evasin subclass, a novel chemokine-binding mechanism for this subclass, and provide insights into how such evasins may be modified to enhance binding and potency. This will be of broad significance to the field of evasins, chemokines and inflammation.

Concerns:

Manuscript text.

1. The abstract states that " mutations to alleviate this negative selection yielded broad-spectrum chemokine inhibitors". Inhibition of chemokine function was only demonstrated for CCL2 and CCL7. To justify this statement inhibition assays against several other chemokines would need to be performed.
2. The introduction states: " Moreover, modification of the chemokine interface of the class A3 evasin yielded engineered evasins with broad spectrum recognition and inhibition of CC chemokines". This is misleading. EVA-A is already known to bind >20 chemokines (see references 11, 16) - it is already "broad-spectrum". The mutations to alleviate the negative selection appeared to reduce binding to chemokines that do not have an aromatic residue at the CC+1 site (Fig. 4a). The mutations appear to be reducing the spectrum of binding rather than increasing it. The authors should modify this statement and fully describe the published binding properties of EVA-A.
3. The results section state that "Moreover, we found that mutation of Cys22 and Cys51 to Ser, giving EVA-A(C8), resulted in significantly diminished binding affinity (or loss of measurable binding) to most chemokines (Fig. 3c)." It is unclear how significance was assessed (see comments below on Fig 3c below). The figure should be revised with assessment of statistical significance to justify this statement.

4. The results section state state: "Consistent with their increased affinity for chemokines with aromatic CC+1 residues, both the Y44A and L39P mutants exhibited significant enhancements in their potencies for inhibition of the same chemokines (Fig. 4b)". Once again, it is unclear how significance was assessed (see comments below on Fig 3c below). The figures should be revised with assessment of statistical significance to justify this statement.

Figures:

Fig. 1b. As the phylogenetic tree is based on sequences in reference 11, to provide concordance, and indicate development of the field, the tree-tips should be labelled as described in reference 11. It will then become clear that A0A0C9S461, A0A023FFD0, and A0A023FT45 are the previously characterised proteins AAM01/E1243, EV991 and EV985 respectively.

Fig. 1d, 3c, f., Fig. 4a, 4d. It is unclear what these bar diagrams show - is it mean / standard error/standard deviation? It is preferable to use a box-whisker plot, showing median and interquartile ranges. Asterisks are traditionally used to show statistical significance but are being used in these figures to show $K_d > 0.1 \text{ nM}$. This is confusing as the y-axis ranges from 0.1 to 100 nM, and individual data points for any $K_d > 0.1 \text{ nM}$ should have been visible. They are not and the authors should address this discrepancy. These figures need revising with all data points clearly indicated. The y-axis order is unorthodox and should start with 0 at the bottom and increase towards the top. Statistical significance of differences in K_d should be calculated and indicated in the figures as appropriate. It is unclear if the individual data points shown in these figures are technical replicates or come from independent experiments (e.g. performed on a different day), and this needs to be clarified in the figure legend. Additionally, Fig.4A has the letters " T L L ..." interposed between top and bottom panels, and "F Y F.." in the top panel. These appear to be the CC+1 residues and should be explained as such in the figure legend.

Methodology:

1. The SPR methods lack the following details: type of signal referencing, a version and name of the software, kinetic model used for fitting.
2. In general for SPR experiments the signal is proportional to a molecule's mass bound to a chip. During the experiments EVA-A was immobilised on the chip surface "to obtain the response unit of 100-150 (RU)". Considering the molecular weights, (EVA-A is $\sim 14 \text{ kDa}$ and CC-chemokines are 7-9 kDa), in the improbable case that 100% of EVA-A molecules are accessible for binding, the signal should be in the range of $\sim 50\text{-}100 \text{ RU}$. However, reported signals often go higher than 200 RU. The authors should comment on this observation.
3. According to the good practice of kinetic measurements, the measured K_d value should be in the centre of the analyte's concentration range (ideally, $0.1 \times K_d - 10 \times K_d$). Although the range used in the

manuscript is 32.5-500 nM, approximately the half of reported constants are lower than 10 nM. That could mask important kinetic information and undermine reliability of the measured K_d . The authors should adjust the concentration range for tight binders during SPR experiments.

4. The authors used single-cycle kinetic experiments to determine K_d values. However, only in few instances data are shown. Moreover, no fitting to the theoretical model is shown what makes impossible to assess accuracy of the fit and K_d estimation. Authors should provide representative single-cycle kinetic sensograms for each measured evasin/chemokine couple and overlay them with a theoretical fit. Ideally, residuals and calculated kinetic constants should be provided as well.

5. Chemokines are known to bind a wide range of glycans which is crucial for their biological activity. In SPR experiments carboxymethyl dextran added to the running buffer and the matrix of the used chip consists of modified dextran. Authors should demonstrate that presence of these glycans doesn't interfere with experimental results.

6. EVA-A was purified by RP-HPLC and analysed by LC-MS. However, no details about these methods are provided.

Reviewer #2 (Remarks to the Author):

Review manuscript Nature Communications

Titel: Engineering Broad-spectrum Inhibitors of Inflammatory Chemokines from a New Family of Tick Evasins

Authors: Shankar Raj Devkota et al.

In this study, the authors described a novel class of evasins, A3 evasins, that is unique to the tick genus *Amblyomma* and distinguished from "classical" class A1 evasins by an additional disulfide-bonded pair of cysteine residues near the chemokine recognition interface. The authors evaluated the structure-activity relationship of this new class of evasins by using various techniques, such as SPR, molecular modelling, X-ray crystallography and cell-based assays. This novel class of evasins can be used for the development of broad-spectrum chemokine targeting polypeptides. From a biological point of view, it may help to arrange and/or confirm the different tick species phylogenetically.

Overall this is a relevant study that revealed a novel evasin class. The structure-function analysis has been investigated thoroughly. The methods used answer the research question and the results support the conclusions.

Minor comments:

Please, be consistent with the names of the classes: A1, A, A3, especially the use of A and A3. This is confusing.

The dots in figure 1B, representing the various tick species are hardly visible. It would be good to make these larger (and other colours).

Major comments:

The authors announce in the title that have uncovered a novel family of tick evasins. However, in the rest of the manuscript it appears to be a new class within the A class evasins, which only bind CC chemokines just like the other A class evasins found in other tick genera.

Apparently, the additional disulfide bond in A3 evasins does not cause a more rigid structure of the protein itself. When they bind to CC chemokines the structure of the A3 evasin becomes more rigid. Free A1 evasins are less flexible compared to free A3 evasins. Does this difference in flexibility: free and bound to a chemokine of interest also affect the affinity, an equilibrium of K_{on} and K_{off} ?

What kind of fitting has been used for SPR experiments? This is not mentioned anywhere.'

Do *Amblyomma* ticks prefer other host species than ticks that possess A1 evasins? If so, do these hosts have another chemokine expression pattern? A few sentences with some discussion about difference in hosts preference (and the need for different chemokine-targeting proteins to feed successfully) of various tick genera is lacking.

Reviewer #3 (Remarks to the Author):

This paper describes a new and large class of high affinity CC chemokine-binding evasins in tick saliva. The authors have taken the research from molecular discovery to crystal structure defining the precise binding interactions with chemokine ligands. They then went further to engineer evasin mutants that alter the chemokine specificity, suggesting that a toolbox of evasins with desirable and precise chemokine binding activity might be developed for clinical translation.

The study extends previous seminal work by Proudfoot et al identifying two other classes of evasins. Overall, the work is rigorously performed and the results are the first description of evasins of this type. Beyond the fascinating biological significance of ticks producing broad spectrum anti-inflammatory agents in their saliva presumably to extend feeding time, the work has potential translational significance for development of novel types of anti-inflammatory agents. Pharma has not picked up on

this previously but perhaps this paper will be the needed catalyst to prevent further evasion of evasins. There are a number of areas where the paper might be improved, all minor. There is no in vivo analysis of evasin activity but given the authors' decision to focus so strongly and well on the structural biology, I don't see this as a requirement for this paper.

Specific comments:

1. The specific chemokines or sets of chemokines bound by class A1 and A2, as well as class B evasins, or by archetypical members of each of these evasin classes, should be listed in the introduction so that the reader can contrast these with the chemokine selectivity of class A3 evasins.
2. Related to the previous question. Evasin-4 (class A1) binds 18 different CC-chemokines, therefore, it appears that the broad CC-chemokine selectivity is not unique to class A3 evasins. This should be mentioned in the discussion. Furthermore, the authors should briefly discuss the evolutionary and physiological meaning of this new family of evasins for ticks. Is it possible that class A3 evolved from evasin-4? Why Amblyomma ticks express different evasins with apparently redundant functionality? Which do the author think is the physiological advantage that these new class A3 evasins provide to these ticks?
3. The sequence alignment in supplementary Figure 1 should include at least one class A1 and one class A2 evasin for reference. Similarly, supplementary Figure 2 should include the sequence of a class A3 evasin.
4. Since this is the first manuscript on this new class of evasins, I think the authors should include as supplementary a Coomassie gel showing the purification of EVA-A as well as information about the yield of the obtained recombinant protein, so that the reader can form a better idea about the purity and glycosylation state of this protein.
5. It is mentioned in the text but there is no evidence provided about the inability of EVA-A to interact with XCL1 or CX3CL1. These SPR sensorgrams should be included in supplementary figure 4.
6. The authors provide a very comprehensive analysis of the binding activity of EVA-A but just a partial analysis of its functional properties (only 5 chemokines are tested in functional assays). There are plenty of examples in the literature that show that ligand binding does not always translate into a functional effect. For instance, evasin-4 is known to bind CCL1, CCL19, CCL21 and CCL25 but it does not inhibit cell migration induced by these chemokines. Therefore, it remains possible that a functional screening

would reveal that the chemokine inhibitory selectivity of EVA-A is much more limited. This should be clearly stated and discussed in the manuscript.

7. The chemokine concentrations used in Fig 1C should be indicated in the figure or the legend.

8. In page 6, the text reads that the K_d shown in Fig 1C corresponds to the dissociation constant but the legend of Fig 1C, as well as other parts of the text and legends of other figures, refer to this K_d as the binding affinity. Dissociation and binding affinity constants are different parameters so please clarify. Also, the legends of figures 1, 3 and 4 indicate that $*K_d$ value > 0.1 nM but this should be $*K_d$ value < 0.1 nM, please correct.

9. There is a sequence of letters in gray above the bars in Figure 1D that is not explained in the text or the figure legend. It was not until I reached Figure 3 that I was able to figure out that these letters indicated the amino acid residue at the CC+1 position for each chemokine. These letters should be moved to Figure 3C, where the authors first mentioned them. Also, there are two inaccuracies. CC+1 of CCL22 is R, not L, and CC+1 of CCL23 is I not R.

10. Figure 1E should include a negative control, a CC-chemokine not bound by EVA-A. Also, I suggest the authors explain a little bit better in the Results what type of functional assay is shown in this figure so that non-chemokine experts among the readership of Nature Communications may understand the rationale behind these assays.

11. Also, what was the chemokine concentration tested in Figures 1E and 4B.

12. In general, all figure legends should be revised to include information about what the different graphs show. Are these means \pm SD? SEM? Are these from one representative experiment? From multiple experiments combined? What do the dots in Figure 3C, Figure 4a, and others indicate? Which statistical analysis was applied in each case? This vital information is missing in all figure legends.

13. The legend of Figure 2 is missing the description of panel D. The current description under D actually corresponds to panel E. Please correct. Also, although I agree with the authors that overall, there are no major effects on the chemokine affinity of the N- or C-terminally truncated EVA-A, there seems to be some potentially interesting differences in the affinity of particular chemokines. However, this is very difficult to interpret because the authors do not show a statistical analysis of these results. Please apply the proper statistical analysis to Figure 2E.

14. In page 8 last paragraph the author say that "In all CC chemokines, the residue immediately following the CC motif (here designated the "CC+1" residue) is a large hydrophobic residue,...". This statement is inaccurate. As shown in the sequence of letters in Figure 1D I mentioned and asked to revise in a previous point, CC+1 of CCL24 is M, CC+1 of CCL27 and CCL15 is T, and CC+1 of CCL22 is R. Methionine (M), threonine (T) and arginine (R) are not large hydrophobic residues. Therefore, this statement should be corrected. This is very important because this hydrophobic CC+1 position is subsequently pointed out by the authors as a vital site for the chemokine interaction with EVA-A. These 4 chemokines do not have a large hydrophobic residue at CC+1 and yet, EVA-A can clearly bind them while it fails to bind CCL2 or CCL20, which do contain a large hydrophobic residue at CC+1. How do the authors explain this?

15. One of the major conclusions of the paper is that EVA-A displays a preference for chemokines with aliphatic residues at position CC+1. Based on the increase in the binding affinity of EVA-A for CCL7 and CCL11 mutants where the CC+1 aromatic residue was mutated to Ala, the authors conclude that EVA-A has evolved to energetically disfavored (negative selection) chemokines with aromatic residues at CC+1. This is framed as a molecular explanation for the chemokine selectivity of EVA-A. However, although this might be true in some cases, in my opinion, I do not think the authors have enough evidence to present this as a general mechanism that explain the ligand selectivity of EVA-A. Figure 1D shows that EVA-A binds 9 chemokines with an affinity constant of 1 nM or lower. Of these 9, 4 are chemokines that present an aromatic residue at position CC+1: CCL3, CCL5, CCL13 and CCL14. Furthermore, Fig 1E shows that EVA-A is a very effective blocker of CCL3. Therefore, it seems that EVA-A has no problem whatsoever in binding and inhibiting chemokines with aromatic amino acids at CC+1. Based on these observations, I suggest the authors should revise and tone down their interpretation of the impact of the chemokine CC+1 residue on the binding to the hydrophobic pocket of EVA-A.

Reviewer #1 Comments

Concerns

1. The abstract states that " mutations to alleviate this negative selection yielded broad-spectrum chemokine inhibitors". Inhibition of chemokine function was only demonstrated for CCL2 and CCL7. To justify this statement inhibition assays against several other chemokines would need to be performed.

We have now performed inhibition assays by measuring phosphorylation of extracellular signal-regulated kinase 1/2 (ERK), in the presence and absence of EVA-A (wild type, Y44A and L39P), for nine CC chemokines: CCL2, CCL3, CCL5, CCL7, CCL8, CCL13, CCL14, CCL15 and CCL23 (**Supplementary Fig. 8** and cited in the manuscript page 6 and 11). For all these chemokines except CCL2, wild type EVA-A inhibited receptor activation. The two mutants inhibited all nine chemokines.

2. The introduction states: " Moreover, modification of the chemokine interface of the class A3 evasin yielded engineered evasins with broad spectrum recognition and inhibition of CC chemokines". This is misleading. EVA-A is already known to bind >20 chemokines (see references 11, 16) - it is already "broad-spectrum". The mutations to alleviate the negative selection appeared to reduce binding to chemokines that do not have an aromatic residue at the CC+1 site (Fig. 4a). The mutations appear to be reducing the spectrum of binding rather than increasing it. The authors should modify this statement and fully describe the published binding properties of EVA-A.

We have modified this statement to read: "Moreover, modification of the chemokine interface of the class A3 evasin yielded engineered evasins with modified selectivity amongst CC chemokines."

In addition, we have compared the K_d values for EVA-A that we determined with those reported by Alenazi et al. 2018 (now ref 17), presented this comparison in Supplementary Fig. 6, and added the statement in the manuscript (page 6): "EVA-A was previously found to bind many of the same chemokines, generally with similar affinities (**Supplementary Fig. 6**). Although statistical comparison of affinities is not possible because only a single K_d value was reported¹⁷, some apparent affinity differences could potentially be attributed to differences in expression systems, post-translational modifications, purity, biophysical methods and/or solution conditions for binding experiments."

3. The results section state that "Moreover, we found that mutation of Cys22 and Cys51 to Ser, giving EVA-A(C8), resulted in significantly diminished binding affinity (or loss of measurable binding) to most chemokines (Fig. 3c)." It is unclear how significance was assessed (see comments below on Fig 3c below). The figure should be revised with assessment of statistical significance to justify this statement.

We have performed and presented the statistical analyses (Fig 3c). We point out that it is more rigorous to perform such statistical comparisons for pK_d values than for K_d values because the pK_d values conform to a normal (Gaussian) distribution, an assumption of standard statistical tests, whereas the K_d values do not conform to a normal distribution, as explained by Christopoulos (*Trends Pharmacol. Sci.*, 1998, **19**, 351–357). We have edited the figure showing pK_d values (and the figure legends) to clearly indicate the relevant statistics (using asterisks) and to indicate cases in which binding was too weak to measure (#) or too tight to measure precisely (\$, $pK_d > 10$, equivalent to $K_d < 0.1$ nM).

In addition, we have modified this statement to read “resulted in significantly or substantially diminished binding affinity (or loss of measurable binding) to most chemokines”. The addition of “or substantially” is needed because, for several chemokines, the affinity of EVA-A (WT) is too tight to measure ($K_d < 0.1$ nM) but changes into the clearly measurable range for the EVA-A(C8) mutant.

4. The results section states: "Consistent with their increased affinity for chemokines with aromatic CC+1 residues, both the Y44A and L39P mutants exhibited significant enhancements in their potencies for inhibition of the same chemokines (Fig. 4b)". Once again, it is unclear how significance was assessed (see comments below on Fig 3c below). The figures should be revised with assessment of statistical significance to justify this statement.

In this case it is not possible to perform a statistical test because the WT evasin inhibits so weakly that its IC_{50} cannot be measured. Instead, we have edited the statement to read: "Consistent with their increased affinity for chemokines with aromatic CC+1 residues, both the Y44A and L39P mutants exhibited substantial enhancements in their potencies for inhibition of CCL2 and CCL7 (Fig. 4b)".

Figures

1. Fig. 1b. As the phylogenetic tree is based on sequences in reference 11, to provide concordance, and indicate development of the field, the tree-tips should be labelled as described in reference 11. It will then become clear that A0A0C9S461, A0A023FFD0, and A0A023FT45 are the previously characterised proteins AAM01/E1243, EV991 and EV985 respectively.

We have modified Fig. 1b to label several relevant “tree-tips”. We have also added **Supplementary Fig. 2**, showing a larger version of the same figure with all “tree-tips” labelled. We agree that this will assist readers in comparisons to the previous literature.

2. Fig. 1d, 3c, f., Fig. 4a, 4d. It is unclear what these bar diagrams show - is it mean / standard error/standard deviation? It is preferable to use a box-whisker plot, showing median and interquartile ranges. Asterisks are traditionally used to show statistical significance but are being used in these figures to show $K_d > 0.1$ nM. This is confusing

as the y-axis ranges from 0.1 to 100 nM, and individual data points for any $K_d > 0.1$ nM should have been visible. They are not and the authors should address this discrepancy. These figures need revising with all data points clearly indicated. The y-axis order is unorthodox and should start with 0 at the bottom and increase towards the top. Statistical significance of differences in K_d should be calculated and indicated in the figures as appropriate. It is unclear if the individual data points shown in these figures are technical replicates or come from independent experiments (e.g. performed on a different day), and this needs to be clarified in the figure legend. Additionally, Fig.4A has the letters " T L L ..." interposed between top and bottom panels, and "F Y F.." in the top panel. These appear to be the CC+1 residues and should be explained as such in the figure legend.

We have improved the presentation of these data in line with the reviewers' suggestions. This includes, showing all three data points as well as the mean and standard error; showing the median and interquartile ranges would not add any new information and would obscure some data points.

The y-axis of each plot represents the pK_d (negative log of the K_d), which is proportional to binding free energy and therefore the most useful parameter for comparing the strength of binding. We have clarified this by including a pK_d scale on the left and a K_d scale on the right of each graph.

We have now used asterisks to indicate statistical significance, as suggested, and clarified the figure legends to confirm that data points are from independent experiments. The previous use of asterisks (now replace by \$ signs) was actually to indicate cases in which $K_d < 0.1$ nM ($pK_d > 10$), explaining why these points were not visible; the text stating that they were for cases in which $K_d > 0.1$ nM was an error, for which we apologise.

Finally, we have moved the labels of the CC+1 residues to the bottom of Fig 3c and explained these labels in the legend.

Methodology:

1. The SPR methods lack the following details: type of signal referencing, a version and name of the software, kinetic model used for fitting.

All relevant details have now been added in the Methods section.

2. In general for SPR experiments the signal is proportional to a molecule's mass bound to a chip. During the experiments EVA-A was immobilised on the chip surface "to obtain the response unit of 100-150 (RU)". Considering the molecular weights, (EVA-A is ~14kDa and CC-chemokines are 7-9 kDa), in the improbable case that 100% of EVA-A molecules are accessible for binding, the signal should be in the range of ~50-100 RU. However, reported signals often go higher than 200 RU. The authors should comment on this observation.

We apologise for this mistake. We checked the experimental details and realised that the amount of EVA-A immobilised on the chip surface was higher than previously stated (typically 200-250 RU). Thus, the RU changes upon chemokine binding in the experiments (up to 220 RU) are consistent with the relative molecular masses of the chemokines (~8-10 kDa) and the tagged EVA-A (~14 kDa). We thank the reviewer for catching this error. The Methods section has been edited as needed.

3. According to the good practice of kinetic measurements, the measured K_d value should be in the centre of the analyte's concentration range (ideally, $0.1 \times K_d - 10 \times K_d$). Although the range used in the manuscript is 32.5-500 nM, approximately the half of reported constants are lower than 10 nM. That could mask important kinetic information and undermine reliability of the measured K_d . The authors should adjust the concentration range for tight binders during SPR experiments.

The reviewer is correct that lower concentrations would normally be ideal for measuring affinity for the tightest binders. This would be essential if the affinities were being measured under equilibrium conditions. However, in our experiments, the affinities are determined by measuring the association rate constants (k_a) and dissociation rate constants (k_d) then taking the ratio of these ($K_d = k_d / k_a$). Using this approach, the concentrations we used are adequate for measuring affinities over a wide range, as long as the kinetic parameters can be determined.

To verify that the concentrations used in our experiments give the same results as using lower concentrations, we have performed the SPR binding experiments for WT EVA-A and 10 chemokines (3 independent measurements) using lower chemokine concentrations. The two sets of pK_d values are presented below. In most cases, they are the same within experimental error. In a few cases, there are small differences but these do not appear to relate to the chemokine concentrations used and they do not affect the interpretation of the data. The errors are similar in both cases. Finally, we note that several of the chemokines dissociate from EVA-A too slowly to allow accurate determination of the dissociation rate constants (k_d). In such cases, it is not possible to determine the K_d using SPR, irrespective of the chemokine concentration used, but we can establish an upper limit of ($K_d < 0.1$ nM).

Figure. Comparison of the chemokine binding affinities (K_d and pK_d) of EVA-A to representative chemokines, as measured using SPR with different ranges of chemokine concentrations. Grey (chemokine concentrations: 31.25, 62.5, 125, 250 and 500, nM); blue (CCL11 and CCL22 concentrations: 1.23, 3.70, 11.11, 33.33 and 100 nM) and red (CCL3, CCL4, CCL5, CCL13, CCL14, CCL16, CCL17 and CCL18 concentrations: 0.12, 0.37, 1.11, 3.33 and 10 nM). The legend lists the highest concentration used. \$, $K_d < 0.1$ nM; * $p < 0.05$, ** $p < 0.01$, *** $p < 0.001$, **** $p < 0.0001$, versus EVA-A binding to chemokine at 31.25-500 nM (two-tailed t-test with Holm-Šídák correction for multiple comparisons).

- The authors used single-cycle kinetic experiments to determine K_d values. However, only in few instances data are shown. Moreover, no fitting to the theoretical model is shown what makes impossible to assess accuracy of the fit and K_d estimation. Authors should provide representative single-cycle kinetic sensograms for each measured evasin/chemokine couple and overlay them with a theoretical fit. Ideally, residuals and calculated kinetic constants should be provided as well.

We have now shown (Fig. 1C, Supplementary Figs 11-17) one representative single-cycle kinetic sensorgram for each evasin-chemokine pair, including fits to the kinetic binding model. In addition, we have added Supplementary Tables 1-9, listing the k_a , k_d and pK_d values (and their standard errors), and the corresponding K_d values, for each evasin-chemokine pair.

- Chemokines are known to bind a wide range of glycans which is crucial for their biological activity. In SPR experiments carboxymethyl dextran added to the running buffer and the matrix of the used chip consists of modified dextran. Authors should demonstrate that presence of these glycans doesn't interfere with experimental results.

To check whether carboxymethyl dextran had any effect on the measured parameters, we have now run a set of SPR experiments in the absence of carboxymethyl dextran. Derived pK_d values are compared below to those measured with carboxymethyl

dextran in the buffer. Differences are negligible (except for a small significant difference for CCL4) and have no effect on the interpretation of our data.

Figure. Comparison of the chemokine binding affinities (K_d and pK_d) of EVA-A to representative chemokines in two different buffers. Grey (buffer with carboxymethyl dextran) and red (buffer without carboxymethyl dextran). \$, $K_d < 0.1$ nM; * $p < 0.05$, ** $p < 0.01$, *** $p < 0.001$, **** $p < 0.0001$ (two-tailed t-test with Holm-Šídák correction for multiple comparisons)

- EVA-A was purified by RP-HPLC and analysed by LC-MS. However, no details about these methods are provided.

We apologise for this omission. The details have been added to the Methods section (pages 20-21).

Reviewer #2 Comments

General Comment

From a biological point of view, it may help to arrange and/or confirm the different tick species phylogenetically.

In Supplementary Fig 1 and Fig 3 (previously Supplementary Fig 2), we have now added phylogenetic trees (Supplementary Fig 1b and Fig 3b) to show the relationships between the sequences listed. We have also inserted notation to indicate which species each sequence is derived from.

Minor Comments

1. Please, be consistent with the names of the classes: A1, A, A3, especially the use of A and A3. This is confusing.

We have edited the manuscript to try and be more consistent. We use “class A” to refer to the whole protein family and “class A1/A2/A3” to refer to the different groups within the class A evasin family.

2. The dots in figure 1B, representing the various tick species are hardly visible. It would be good to make these larger (and other colours).

We have clarified this figure, as noted above (Reviewer #1, Figures, point #1).

Major Comments

1. The authors announce in the title that have uncovered a novel family of tick evasins. However, in the rest of the manuscript it appears to be a new class within the A class evasins, which only bind CC chemokines just like the other A class evasins found in other tick genera.

We have toned down this claim, as suggested, both in the title and throughout the manuscript.

2. Apparently, the additional disulfide bond in A3 evasins does not cause a more rigid structure of the protein itself. When they bind to CC chemokines the structure of the A3 evasin becomes more rigid. Free A1 evasins are less flexible compared to free A3 evasins. Does this difference in flexibility: free and bound to a chemokine of interest also affect the affinity, an equilibrium of K_{on} and K_{off} ?

It is difficult to answer this question unequivocally. Generally, a decrease in flexibility upon binding is taken to indicate that tighter (more well-defined) interactions (hydrogen bonds, hydrophobic interactions, etc.) are being formed, which typically corresponds to higher affinity. However, these well-defined interactions have an entropic cost. To acknowledge this entropy-enthalpy compensation, we have added the following sentence (page 13). “While the reduced flexibility of EVA-A suggests a greater entropic cost to forming these interactions (compared to EVA-P), clearly the enthalpic advantage of forming the interactions outweighs this cost, otherwise they would not form.”

For the reviewer’s reference, we have included below a comparison between the $\log(k_a)$, $\log(k_d)$ and pK_d values for EVA-A and EVA-P. We note that three chemokines (CCL7, CCL8 and CCL11) bind more tightly to EVA-P than to EVA-A. In all three cases, the tighter binding is a result of slower dissociation. On the other hand, EVA-A binds more tightly to five chemokines (CCL13, CCL14, CCL16, CCL17, CCL18). In all of these cases, EVA-A has slower dissociation than EVA-P. In four of the cases (all except CCL18), EVA-A also has faster association than EVA-P. Relating these differences to protein flexibility would probably require molecular dynamics simulations (or experimental measurements of flexibility) for the various chemokine complexes, which would be a substantial undertaking.

Figure. Comparison of the binding kinetics parameters, $\log(k_a)$ and $\log(k_d)$, and the equilibrium dissociation constant (pK_d) between EVA-A and EVA-P.

3. What kind of fitting has been used for SPR experiments? This is not mentioned anywhere.'

We apologise for this omission. These details have now been added to the Methods section.

4. Do Amblyomma ticks prefer other host species than ticks that possess A1 evasins? If so, do these hosts have another chemokine expression pattern? A few sentences with some discussion about difference in hosts preference (and the need for different

chemokine-targeting proteins to feed successfully) of various tick genera is lacking.

Although we can only speculate about the evolutionary advantages of class A3 evasins, we have added the following paragraph (page 15) to address this question.

“The emergence of class A3 evasins in American *Amblyomma* ticks suggests that class A3 evasins confer a competitive advantage for these species. Considering that *Amblyomma* ticks infest a wide variety of species²², it is possible that the broad chemokine binding profiles of class A3 evasins facilitate the spread of ticks amongst different host species. On the other hand, ticks from other genera, such as *Rhipicephalus*, also infest a variety of species,²³⁻²⁵ so the occurrence of class A3 evasins in *Amblyomma* species may not be a dominant factor in controlling host compatibility. Indeed, an alternative strategy to achieve broad chemokine inhibition is simply to secrete a cocktail of evasins with different chemokine selectivities, as discussed previously for the species *R. pulchellus*¹¹.”

Reviewer #3 Comments

Specific Comments (all minor)

1. The specific chemokines or sets of chemokines bound by class A1 and A2, as well as class B evasins, or by archetypical members of each of these evasin classes, should be listed in the introduction so that the reader can contrast these with the chemokine selectivity of class A3 evasins.

We have added the following summary of this information to the introduction (page 4): “For example, the class A evasins EVA-1 and EVA-4 bind to four and 17 human CC chemokines, respectively¹¹, whereas the class B evasin EVA-3 binds to six CXC chemokines¹².”

2. Related to the previous question. Evasin-4 (class A1) binds 18 different CC-chemokines, therefore, it appears that the broad CC-chemokine selectivity is not unique to class A3 evasins. This should be mentioned in the discussion. Furthermore, the authors should briefly discuss the evolutionary and physiological meaning of this new family of evasins for ticks. Is it possible that class A3 evolved from evasin-4? Why *Amblyomma* ticks express different evasins with apparently redundant functionality? Which do the author think is the physiological advantage that these new class A3 evasins provide to these ticks?

We have acknowledged the relatively broad CC chemokine selectivity of EVA-4 by adding the following statement on page 6: “Whereas EVA-4 is also a broad-spectrum binder to CC chemokines¹¹, EVA-4 and EVA-A exhibit quite distinct chemokine affinity profiles (**Supplementary Fig. 7**.)” Supplementary Fig. 7 is a new figure to support this statement.

Since class A3 evasins are only found in *Amblyomma* ticks, it is clear that they did not evolve directly from either EVA-1 or EVA-4, which are found in a species from a different genus (*Rhipicephalus*). For this reason, we concluded (page 5): “Thus, class A3 evasins constitute a specialised evasin family that has evolved after divergence of *Amblyomma* from other genera of ticks.”

Regarding the possible physiological advantages of class A3 evasins and the reasons for expressing multiple chemokines with overlapping functionality, please see our response to Reviewer #2, major comment #4.

3. The sequence alignment in supplementary Figure 1 should include at least one class A1 and one class A2 evasin for reference. Similarly, supplementary Figure 2 should

include the sequence of a class A3 evasin.

These have been added, as suggested. Note that the figure previously labelled **Supplementary Fig. 2** is now **Supplementary Fig. 3**.

4. Since this is the first manuscript on this new class of evasins, I think the authors should include as supplementary a Coomassie gel showing the purification of EVA-A as well as information about the yield of the obtained recombinant protein, so that the reader can form a better idea about the purity and glycosylation state of this protein.

A Coomassie gel has been added (Supplementary Fig. 4a). The typically yield (~0.5-0.8 mg per L of culture) has been added to the Methods section.

5. It is mentioned in the text but there is no evidence provided about the inability of EVA-A to interact with XCL1 or CX3CL1. These SPR sensorgrams should be included in supplementary figure 4.

The SPR sensorgrams for XCL1 and CX₃CL1 for were omitted in error. They have now been added to Supplementary Figure 5.

6. The authors provide a very comprehensive analysis of the binding activity of EVA-A but just a partial analysis of its functional properties (only 5 chemokines are tested in functional assays). There are plenty of examples in the literature that show that ligand binding does not always translate into a functional effect. For instance, evasin-4 is known to bind CCL1, CCL19, CCL21 and CCL25 but it does not inhibit cell migration induced by these chemokines. Therefore, it remains possible that a functional screening would reveal that the chemokine inhibitory selectivity of EVA-A is much more limited. This should be clearly stated and discussed in the manuscript.

We have performed additional functional screening (**Supplementary Fig. 8**). EVA-A inhibited receptor activation by all chemokines tested. We note that, in our hands, EVA-4 is either a very weak binder or does not bind measurably to the chemokines listed by the reviewer (see **Supplementary Fig. 7**). Considering that class A evasins bind to the regions of CC chemokines required for receptor binding and activation, tight binding (specially with slow dissociation) is generally expected to result in functional inhibition, as we have observed here.

7. The chemokine concentrations used in Fig 1C should be indicated in the figure or the legend.

This information has been added in the Figure legend.

8. In page 6, the text reads that the K_d shown in Fig 1C corresponds to the dissociation constant but the legend of Fig 1C, as well as other parts of the text and legends of other figures, refer to this K_d as the binding affinity. Dissociation and binding affinity constants are different parameters so please clarify. Also, the legends of figures 1, 3 and 4 indicate that $*K_d$ value > 0.1 nM but this should be $*K_d$ value < 0.1 nM, please correct.

The K_d is the dissociation equilibrium constant and is the parameter normally used to define binding affinity (lower K_d means higher affinity).

We have corrected the error to indicate cases in which K_d value < 0.1 nM.

9. There is a sequence of letters in gray above the bars in Figure 1D that is not explained in the text or the figure legend. It was not until I reached Figure 3 that I was able to figure out that these letters indicated the amino acid residue at the CC+1 position for each chemokine. These letters should be moved to Figure 3C, where the authors first mentioned them. Also, there are two inaccuracies. CC+1 of CCL22 is R, not L, and CC+1 of CCL23 is I not R.

We have moved the CC+1 labels to Fig 3c, as suggested and fixed the two inaccuracies (impressively detailed work by the reviewer!).

10. Figure 1E should include a negative control, a CC-chemokine not bound by EVA-A. Also, I suggest the authors explain a little bit better in the Results what type of functional assay is shown in this figure so that non-chemokine experts among the readership of Nature Communications may understand the rationale behind these assays.

We have clarified our description of the functional assay in the Fig 1e legend.

Regarding a negative control, we assume the reviewer is asking for evidence that a non-binding chemokine is also not inhibited by EVA-A. We have therefore added data for CCL2 to Fig 1e; note that these are the same data shown in Fig 4b. They clearly show that the activity of CCL2 is not inhibited by EVA-A.

11. Also, what was the chemokine concentration tested in Figures 1E and 4B.

This information has been added to the figure legends.

12. In general, all figure legends should be revised to include information about what the different graphs show. Are these means \pm SD? SEM? Are these from one representative experiment? From multiple experiments combined? What do the dots in Figure 3C, Figure 4a, and others indicate? Which statistical analysis was applied in each case? This vital information is missing in all figure legends.

As noted above, all figures and legends have been improved as requested.

13. The legend of Figure 2 is missing the description of panel D. The current description under D actually corresponds to panel E. Please correct. Also, although I agree with the authors that overall, there are no major effects on the chemokine affinity of the N- or C-terminally truncated EVA-A, there seems to be some potentially interesting differences in the affinity of particular chemokines. However, this is very difficult to interpret because the authors do not show a statistical analysis of these results. Please apply the proper statistical analysis to Figure 2E.

Apologies for the omitted legend for Fig 2d. We have added this.

We have applied the statistical analysis to the data in Fig 2e. Indeed, we found that some of the changes are significant, albeit fairly small. We have edited the relevant text (page 9) to read: “with only small (albeit some significant) losses of affinity...”

14. In page 8 last paragraph the author say that “In all CC chemokines, the residue immediately following the CC motif (here designated the “CC+1” residue) is a large hydrophobic residue,...”. This statement is inaccurate. As shown in the sequence of letters in Figure 1D I mentioned and asked to revise in a previous point, CC+1 of CCL24 is M, CC+1 of CCL27 and CCL15 is T, and CC+1 of CCL22 is R. Methionine (M), threonine (T) and arginine (R) are not large hydrophobic residues. Therefore, this statement should be corrected. This is very important because this hydrophobic CC+1 position is subsequently pointed out by the authors as a vital site for the chemokine interaction with EVA-A. These 4 chemokines do not have a large hydrophobic residue at CC+1 and yet, EVA-A can clearly bind them while it fails to bind CCL2 or CCL20, which do contain a large hydrophobic residue at CC+1. How do the authors explain this?

Actually, Met, Arg and Thr all have substantial hydrophobic character. Met is predominantly hydrophobic, whereas both Arg and Thr have at least three hydrophobic carbon atoms, albeit with additional polar functional groups. We have edited the relevant section to read as follows: “In all CC chemokines, the residue immediately following the CC motif (here designated the “CC+1” residue) has substantial hydrophobic character (at least three hydrophobic carbon atoms) and the potential to form hydrophobic interactions”.

Regarding the additional point that EVA-A fails to bind CCL2 or CCL20, it seems that there are other features of these chemokines that disfavour tight binding to EVA-A.

15. One of the major conclusions of the paper is that EVA-A displays a preference for chemokines with aliphatic residues at position CC+1. Based on the increase in the binding affinity of EVA-A for CCL7 and CCL11 mutants where the CC+1 aromatic residue was mutated to Ala, the authors conclude that EVA-A has evolved to energetically disfavored (negative selection) chemokines with aromatic residues at CC+1. This is framed as a molecular explanation for the chemokine selectivity of EVA-A. However, although this might be true in some cases, in my opinion, I do not think the authors have enough evidence to present this as a general mechanism that explain the ligand selectivity of EVA-A. Figure 1D shows that EVA-A binds 9 chemokines with an affinity constant of 1 nM or lower. Of these 9, 4 are chemokines that present an aromatic residue at position CC+1: CCL3, CCL5, CCL13 and CCL14. Furthermore, Fig 1E shows that EVA-A is a very effective blocker of CCL3. Therefore, it seems that EVA-A has no problem whatsoever in binding and inhibiting chemokines with aromatic amino acids at CC+1. Based on these observations, I suggest the authors should revise and tone down their interpretation of the impact of the chemokine CC+1 residue on the binding to the hydrophobic pocket of EVA-A.

We take the reviewer's point that this effect may not be as general as we initially indicated. We have edited the text accordingly. Specifically, we have made the following changes.

Pages 11-13

- We have deleted the sentence: “Moreover, we recognised that chemokines with aliphatic CC+1 residues tend to have higher affinity than those with aromatic CC+1 residues (Fig. 1d).”
- Previous text: “interactions of the aromatic CC+1 residues had imposed an energetic penalty”
- Edited text: “interactions of the aromatic CC+1 residues in these chemokines had imposed an energetic penalty”
- Previous text: “the preference of EVA-A for chemokines with aliphatic CC+1 residues is because aromatic residues are energetically disfavoured”
- Edited text: “the preference of EVA-A for CCL16 over CCL7 and CCL11, is because the aromatic residues in the latter two chemokines are energetically disfavoured”

- Previous heading: “Engineering broad-spectrum binding by removal of negative selection”
- Edited heading: “Modifying chemokine selectivity by removal of negative selection”
- Previous text: “Considering the above finding that the interactions of aromatic CC+1 residues with the hydrophobic pocket impose an energetic penalty against high affinity binding, we postulated that EVA-A could potentially be converted into a broad-spectrum CC chemokine binder by alleviating this negative selection.”
- Edited text: “Considering the above finding that the interactions of aromatic CC+1 residues of CCL7 and CCL11 with the hydrophobic pocket impose an energetic penalty against high affinity binding, we postulated that the affinity of EVA-A for these (and possibly other) chemokines could potentially be enhanced by alleviating this negative selection.”
- Previous text: “In the case of L39P, the affinity decreases were only small so this mutant bound to 21 of the 24 available CC chemokines with affinities tighter than 100 nM, and can be considered a broad-spectrum CC chemokine binder.”
- Edited text: “In the case of L39P, the affinity decreases were only small so this mutant bound to 21 of the 24 available CC chemokines with affinities tighter than 100 nM, compared to 17 chemokines for wild type EVA-A.”
- Previous text: “This indicated that the evasin mutants no longer display strong negative selection against binding to chemokines with aromatic CC+1 residues. Thus, the broad-spectrum chemokine binding and inhibition of EVA-A(L39P) resulted from the removal of negative selection.”
- Edited text: “This indicated that the evasin mutants no longer display strong negative selection against binding to the aromatic CC+1 residues of CCL7 and CCL11. Thus, the increased breadth of chemokine binding by EVA-A(L39P) resulted, at least in part, from the removal of negative selection.”

Page 17-18

- Previous text: “The detailed structure of the hydrophobic pocket may enable a class A3 evasin to distinguish between chemokines with different CC+1 residues”
- Edited text: “The detailed structure of the hydrophobic pocket may enable a class A3 evasin to exhibit preferences for chemokines with different CC+1 residues”
- Previous text: “may accommodate a wide variety of CC+1 residues, enabling broad-spectrum chemokine binding, as observed here for the EVA-A(L39P) mutant”
- Edited text: “may accommodate a relatively wide variety of CC+1 residues, as observed here for the EVA-A(L39P) mutant”

- Previous text: “the preference of EVA-A for chemokines with aliphatic CC+1 residues results from negative selection against those with aromatic CC+1 residues ”
- Edited text: “the preference of EVA-A for chemokines with aliphatic CC+1 residues results from negative selection against some chemokines with aromatic CC+1 residues”

- Previous text: “enabled us to reprogram EVA-A as a promiscuous CC chemokine binder”
- Edited text: “enabled us to reprogram EVA-A as a more promiscuous CC chemokine binder”

REVIEWERS' COMMENTS

Reviewer #1 (Remarks to the Author):

The authors have made several improvements to the manuscript and figures in accordance with the reviewers' comments and have added new experimental data which justify the conclusions.

Comments

Methods

There is no statistics subsection in the methods. This should contain a description of methods and software used for statistical analyses and concentration-response curve fitting.

Figures:

1d. It is unclear what the figure shows as the summary statistic (presumably mean) and error bars (presumably SE) are not defined in the figure or the legend.

2e. It is unclear what the figure shows as the summary statistic (presumably mean) and error bars (presumably SE) are not defined. Why was a two-way ANOVA rather than a one-way ANOVA used? The authors could use a one-way ANOVA e.g. as in Supplementary Fig. 8, where the only factor is the evasin. For a two-way ANOVA, the two factors and the three hypothesis pairs - which includes the interaction between the two factors - should be explained. ANOVA statistics including p, F, and degrees of freedom should be reported as required in the submission guide for statistical information.

3c. It is unclear what the figure shows as the summary statistic (presumably mean) and error bars (presumably SE) are not defined.

3f. It is unclear what the figure shows as the summary statistic (presumably mean) and error bars (presumably SE) are not defined.

4a. It is unclear what the figure shows as the summary statistic (presumably mean) and error bars (presumably SE) are not defined. Why was a two-way ANOVA rather than a one-way ANOVA used? For a two-way ANOVA, the two factors and the three hypothesis pairs - which includes the interaction between the two factors - should be explained. The authors could use a one-way ANOVA e.g. as in Supplementary Fig. 8, where the only factor is the evasin. ANOVA statistics including p, F, and degrees of freedom should be reported as required in the submission guide for statistical information.

4b. It is unclear what the figure shows as the summary statistic (presumably mean) and error bars (presumably SE) are not defined.

4d. It is unclear what the figure shows as the summary statistic (presumably mean) and error bars (presumably SE) are not defined.

Reviewer #2 (Remarks to the Author):

According to this reviewer, all concerns and feedback have been addressed properly. The manuscript has been improved and there are no more questions or comments.

Reviewer #1 Comments

Methods:

There is no statistics subsection in the methods. This should contain a description of methods and software used for statistical analyses and concentration-response curve fitting.

We have now added the subsection “Quantification and statistical analysis” in the Methods section.

Figures:

1d. It is unclear what the figure shows as the summary statistic (presumably mean) and error bars (presumably SE) are not defined in the figure or the legend.

We have now added the sentence “Data are presented as mean \pm SEM from three independent experiments” in the figure legend.

2e. It is unclear what the figure shows as the summary statistic (presumably mean) and error bars (presumably SE) are not defined. Why was a two-way ANOVA rather than a one-way ANOVA used? The authors could use a one-way ANOVA e.g. as in Supplementary Fig. 8, where the only factor is the evasin. For a two-way ANOVA, the two factors and the three hypothesis pairs - which includes the interaction between the two factors - should be explained. ANOVA statistics including p, F, and degrees of freedom should be reported as required in the submission guide for statistical information.

We have now added the sentence “Data are presented as mean \pm SE from three independent experiments” in the figure legend. As the reviewer suggested, we have performed and presented the statistical analyses using one-way ANOVA. We have updated the figure and figure legend.

3c. It is unclear what the figure shows as the summary statistic (presumably mean) and error bars (presumably SE) are not defined.

We have now added the sentence “Data are presented as mean \pm SEM from three independent experiments” in the figure legend.

3f. It is unclear what the figure shows as the summary statistic (presumably mean) and error bars (presumably SE) are not defined.

We have now added the sentence “Data are presented as mean \pm SEM from three independent experiments” in the figure legend.

4a. It is unclear what the figure shows as the summary statistic (presumably mean) and error bars (presumably SE) are not defined. Why was a two-way ANOVA rather than a one-way ANOVA used? For a two-way ANOVA, the two factors and the three hypothesis pairs - which includes the interaction between the two factors - should be explained. The authors could use a one-way ANOVA e.g. as in Supplementary Fig. 8, where the only factor is the evasin. ANOVA statistics including p, F, and degrees of freedom should be reported as required in the submission guide for statistical information.

We have now added the sentence “Data are presented as mean \pm SEM from three independent experiments” in the figure legend. As the reviewer suggested, we have performed and presented the statistical analyses using one-way ANOVA. We have updated the figure and figure legend.

4b. It is unclear what the figure shows as the summary statistic (presumably mean) and error bars (presumably SE) are not defined.

We have now edited the figure legend to include a more detailed description of the experiment, including the statement “Data are... presented as mean \pm SEM from three independent experiments”.

4d. It is unclear what the figure shows as the summary statistic (presumably mean) and error bars (presumably SE) are not defined.

We have now added the sentence “Data are presented as mean \pm SEM from three independent experiments” in the figure legend.